# Simple and Asymmetric Graph Contrastive Learning without Augmentations

**Teng Xiao**[1*], **Huaisheng Zhu**[1*], **Zhengyu Chen**[2], **Suhang Wang**[1]
[1]The Pennsylvania State University, [2]Zhejiang University,
{tengxiao,hvz5312,szw494}@psu.edu, chenzhengyu@zju.edu.cn

## Abstract

Graph Contrastive Learning (GCL) has shown superior performance in representation learning in graph-structured data. Despite their success, most existing GCL methods rely on prefabricated graph augmentation and homophily assumptions. Thus, they fail to generalize well to heterophilic graphs where connected nodes may have different class labels and dissimilar features. In this paper, we study the problem of conducting contrastive learning on homophilic and heterophilic graphs. We find that we can achieve promising performance simply by considering an asymmetric view of the neighboring nodes. The resulting simple algorithm, Asymmetric Contrastive Learning for Graphs (GraphACL), is easy to implement and does not rely on graph augmentations and homophily assumptions. We provide theoretical and empirical evidence that GraphACL can capture one-hop local neighborhood information and two-hop monophily similarity, which are both important for modeling heterophilic graphs. Experimental results show that the simple GraphACL significantly outperforms state-of-the-art graph contrastive learning and self-supervised learning methods on homophilic and heterophilic graphs. The code of GraphACL is available at https://github.com/tengxiao1/GraphACL.

## 1 Introduction

Contrastive learning has emerged as a promising regime for unsupervised vision representation learning without using annotated labeled data [1]. Recently, graph contrastive learning (GCL) has also been introduced to graph-structured data due to the lack of task-specific node labels [2, 3, 4, 5, 6, 7]. GCL has achieved competitive (or even better) performance on many downstream tasks on graphs compared to its counterparts trained with annotated ground-truth labels [7, 8].

Generally, existing GCL methods can be categorized into two categories. The first contrastive scheme [9, 10, 11, 12] reconstructs the local network structure (i.e., observed edges) to align with traditional network-embedding objectives [13, 14]. Specifically, this scheme treats one-hop or random-walk local neighboring nodes of the target node as positive examples and non-neighboring nodes as negative samples. It then employs contrastive loss functions to minimize the representation distance of positive pairs and maximize the distance of negative pairs [10]. The key motivation behind this scheme is the *explicit* homophily assumption that semantically similar nodes are more likely to be linked than dissimilar ones. Thus, connected nodes should have similar representations in the latent space. However, in heterophilic graphs, connected nodes are not necessarily from the same semantic class [15, 16], and should not be simply pulled together in the latent space. This scheme empirically faces issues with performance in graphs exhibiting heterophily [17, 18].

The second graph contrastive scheme involves graph augmentations [3, 4, 5, 6, 7]. Specifically, it constructs two views through stochastic graph augmentation and then learns representations by contrasting these views based on the information maximization principle [19]. A positive node

---

[*]Equal contribution

37th Conference on Neural Information Processing Systems (NeurIPS 2023).

pair consists of two views resulting from stochastic data augmentation of the same node, while a negative pair might consist of two views of different nodes [8]. This scheme is built based on the idea that augmentations can preserve the semantic nature of samples, i.e. augmented nodes have consistent semantic labels with the same original nodes. However, recent work has shown that graph augmentation methods struggle to achieve good performance on heterophilic graphs [18] as they still *implicitly* rely on homophily [20]. Studies have shown that stochastic augmentation primarily captures the common low-frequency information between two views, while neglecting the high-frequency one [20] (also see Appendix C.4). The latter is known to be more crucial for heterophilic graphs [21, 22]. Given that heterophilic graphs are prevalent across various real-world domains [16, 23], a fundamental and open question naturally arises: *What kind of contrastive learning objectives can learn robust node representations on both homophilic and heterophilic graphs?*

This work is the first to address the aforementioned question without relying on augmentations and homophily assumptions. Instead, we take into account two more generalized insights, as depicted in Figure 1. (i) Even in heterophilic graphs, two nodes of the same semantic class often share a similar one-hop neighborhood context [24, 18]. Thus, capturing this heterophilic one-hop neighborhood context can lead to more discriminative node representations. (ii) While homophily might be minimal or non-existent in heterophilic graphs, another frequently observed

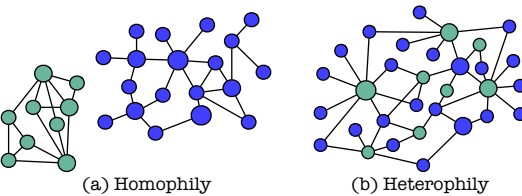

(a) Homophily  (b) Heterophily

Figure 1: The graphs of pure homophily and pure heterophily, where color denotes the semantic class. For both graphs, nodes with a similar one-hop neighborhood context have similar semantic classes and two-hop similarities still exist even without the one-hop homophily.

phenomenon in real-world graphs is monophily [25]. *Monophily* describes situations in real-world graphs where the attributes of a node's friends are likely to be similar to the attributes of that node's other friends [25]. Put another way, "monophily" essentially induces similarities between two-hop neighbors. As noted by [25, 26], the two-hop similarities brought about by monophily can persist even in the total absence of any one-hop similarities that might be suggested by homophily.

To support our motivations, we provide statistics of homophily, two-hop monophily ratios, and neighborhood similarities of various graphs in the real-world in Appendix C. As shown in these statistics, homophilic and heterophilic graphs in the real-world exhibit relatively strong neighborhood similarity calculated based on one-hop neighborhoods compared to homophily. In cases where homophily is weak or non-existent, monophily has been shown to still hold in real-world graphs.

In this work, we focus on exploiting the above insights to design new objectives for better node representations on both homophilic and heterophilic graphs. We propose a simple yet effective framework termed as graph asymmetric contrastive learning (GraphACL) for better node representation learning. Essentially, we are faced with the challenge of simultaneously capturing the one-hop neighborhood context and monophily in the contrastive objective. To solve this challenge, we consider each node to play two roles: the node itself (identity representation) and specific "neighbors" of other nodes (context representation), and thus should be treated differently. GraphACL trains the node identity representation by predicting the context representation of one-hop neighbors through an asymmetric predictor. Intuitively, by enforcing identity representations of two-hop neighbors to reconstruct the same context representation of the same central nodes, GraphACL implicitly makes representations of two-hop neighbors similar and captures the one-hop neighborhood context (Figure 2).

**Our primary technical contributions are: (1)** We propose a *simple*, *effective*, and *intuitive* graph contrastive learning approach which captures one-hop neighborhood context and two-hop monophily similarities in a simple asymmetric learning framework. **(2)** We theoretically analyze the learning behavior and prove that GraphACL is guaranteed to yield good node representations for both homophilic and heterophilic graphs. **(3)** Empirically, we corroborate the effectiveness of GraphACL on 15 graph benchmarks. The results demonstrate that GraphACL can significantly outperform previous GCL methods. The surprising effectiveness of GraphACL shows that in the context of GCL, simpler methods have been underexplored in favor of more elaborate algorithmic contributions.

## 2   Related Work

**Graph Contrastive Learning.** Contrastive methods are central to traditional network-embedding methods [27, 13, 14], but have recently been applied in graph self-supervised learning [9, 28, 10].

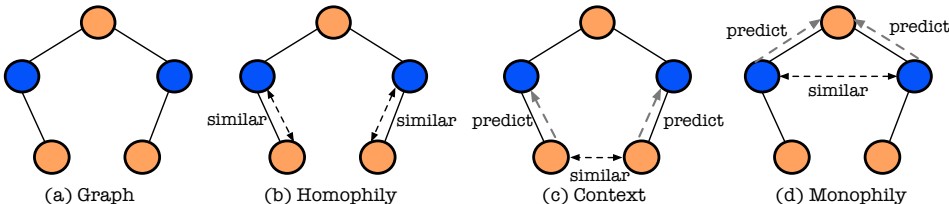

Figure 2: An illustration of various design motivations. (a) The heterophilic graph where the color denotes node's semantic class. (b) Contrastive objectives with the homophily assumption encourage one-hop neighbors to have similar representations. GraphACL simply encourages the node to predict its neighbors, which can implicitly capture neighborhood context (c) and two-hop monophily (d).

The key motivation behind them is the *explicit* homophily assumption that connected nodes belong to the same class and, thus, should be treated as positive pairs in contrastive learning. However, real-world graphs do not always obey the homophily assumption [16], limiting their applicability to heterophilic graphs [18]. Recently, many GCL algorithms with augmentations [2, 3, 4, 5, 6, 29] have been proposed. However, [20] recently proved that GCL with augmentations attempts to manipulate the encoder to capture low-frequency information instead of the high-frequency part and suffers performance degradation in heterophilic graphs [18]. In contrast, we propose a simple asymmetric contrastive learning framework for graphs without augmentations and homophily assumption.

**Heterophilic Graphs.** There are many heterophilic graphs in the real world that exhibit nonho-mophilic properties, such as transaction [30, 31], ecological food [32] and molecular networks [15], where the linked nodes have different characteristics and different class labels. Various graph neural networks (GNNs) [33, 15, 21, 34, 35, 23, 36, 37] have been proposed to achieve higher performance in low homophily settings. They focus on designing advanced GNN architectures and consider the semi-supervised setting with labels. In contrast, we focus on designing the contrastive learning algorithm without labels, not on specific GNN architectures. Recently, DSSL [18] and HGRL [29] have been proposed to conduct self-supervised learning on nonhomophilous graphs by capturing global and high-order information. Specifically, HGRL is based on graph augmentation, and DSSL works by assuming a graph generation process, which may not always hold true for real-world graphs. In contrast, our work provides a novel perspective on graph contrastive learning, which does not make any assumptions about the generation process or rely on any augmentations.

## 3 Preliminaries

**Notations and Problem.** Let $\mathcal{G} = (\mathcal{V}, \mathcal{E})$ be a input graph, where $\mathcal{V} = \{v_1, \ldots, v_{|\mathcal{V}|}\}$ is the set of $|\mathcal{V}|$ nodes and $\mathcal{E}$ is the set of edges. Let $\mathbf{X} = [\mathbf{x}_1, \mathbf{x}_2, \cdots, \mathbf{x}_{|\mathcal{V}|}] \in \mathbb{R}^{|\mathcal{V}| \times D_x}$ be the node attribute matrix, where $\mathbf{x}_i$ is the $D_x$-dimensional feature vector of $v_i$. Each edge $e_{i,j} \in \mathcal{E}$ denotes a link between node $v_i$ and $v_j$. The graph structure can be denoted by an adjacency matrix $\mathbf{A} \in [0, 1]^{|\mathcal{V}| \times |\mathcal{V}|}$ with $\mathbf{A}_{i,j} = 1$ if $e_{i,j} \in \mathcal{E}$, otherwise $\mathbf{A}_{i,j} = 0$. We denote the normalized adjacency matrix $\tilde{\mathbf{A}} = \mathbf{D}^{-1/2}\mathbf{A}\mathbf{D}^{-1/2}$ where $\mathbf{D}$ is the diagonal degree matrix. Let $\mathbf{L} = \mathbf{I} - \tilde{\mathbf{A}}$ be the symmetric normalized graph Laplacian matrix. We also define the unweighted two-hop graph $\mathcal{G}_2$, whose adjacency matrix is $\mathbf{A}_2$. $(\mathbf{A}_2)_{ik} = 1$ if there exists $j$ such that $e_{ij} \in \mathcal{E}$ and $e_{jk} \in \mathcal{E}$. The input graph $\mathcal{G}$ can be denoted as a tuple of matrices $G = (\mathbf{X}, \mathbf{A})$. Our goal is to learn a GNN encoder $f_\theta$ parameterized by $\theta$, such that the representation of node $v$: $\mathbf{v} = f_\theta(G)[v] \in \mathbb{R}^D$, can perform well for downstream tasks.

**Homophily Ratio.** Typically, the graph homophily ratio [33, 15] is defined as the fraction of edges connecting nodes with the same labels, i.e., $\mathcal{H}(\mathcal{G}) = |\{(u,v) : (u,v) \in \mathcal{E} \wedge y_u = y_v\}|/|\mathcal{E}|$. Homophily Graphs have high edge homophily ratio $\mathcal{H}(\mathcal{G}) \to 1$, while heterophilic graphs (i.e., low/weak homophily) correspond to small edge homophily ratio $\mathcal{H}(\mathcal{G}) \to 0$ [15, 16].

**GCL with Representation Smoothing.** This paradigm [13, 9, 28, 12] ensures that local neighboring nodes have similar representations and has also been shown to be equivalent to factorizing graph proximity [38]. Despite minor differences, the objective for this paradigm can be expressed as:

$$\mathcal{L}_{\mathrm{s}} = -\frac{1}{|\mathcal{V}|} \sum_{v \in \mathcal{V}} \frac{1}{|\mathcal{N}(v)|} \sum_{u \in \mathcal{N}(v)} \log \frac{\exp(\mathbf{v}^\top \mathbf{u}/\tau)}{\exp\left(\mathbf{v}^\top \mathbf{u}/\tau\right) + \sum_{v_- \in \mathcal{V}_-} \exp\left(\mathbf{v}^\top \mathbf{v}_-/\tau\right)}. \tag{1}$$

Here $\mathbf{v}$, $\mathbf{u}$ and $\mathbf{v}_-$ are the projected node representations by $f_\theta(G)$ of nodes $v$, $u$ and $v_-$, respectively. $\tau$ is the temperature hyper-parameter. Typically, $\mathcal{N}(v)$ is the positive sample set containing one-hop

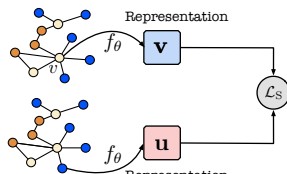
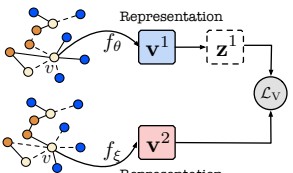
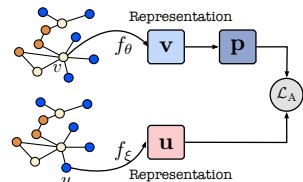

| (a) GCL with Representation Smoothing | (b) GCL with Augmented Views | (c) Graph Asymmetric Contrastive Learning |

Figure 3: Illustration of existing contrastive schemes and GraphACL. (a) forces neighboring nodes to have similar representations based on the homophily assumption. (b) augments the graph and learns the augment-invariant representations of the same node. Our GraphACL in (c) simply reconstructs the neighborhood signal of each node based on an asymmetric predictor without relying on the homophily assumption and augmentation.

local neighborhoods of node $v$, and $\mathcal{V}_-$ is the negative sample set that can be randomly sampled from the node space $\mathcal{V}$ or sampled proportionally to the power $3/4$ of the degree of the node [14]. Figure 3 (a) illustrates this contrastive learning scheme with this explicit homophily assumption.

**GCL with Augmented Views.** This paradigm [4, 20, 39, 3] learns representations by contrasting views based on stochastic augmentations. Specifically, for node $v$, its representation in one augmented view is learned to be close to the representation of the same node $v$ from the other augmented view and far away from the representations of negative samples from other nodes. Given two augmentations $G_1$ and $G_2$ extracted from original graph $G$ in a predefined way, the contrastive objective is as follows:

$$\mathcal{L}_\text{V} = -\frac{1}{|\mathcal{V}|} \sum\nolimits_{v \in \mathcal{V}} \log \frac{\exp(\mathbf{v}^1 \cdot \mathbf{v}^2/\tau)}{\exp\left(\mathbf{v}^1 \cdot \mathbf{v}^2/\tau\right) + \sum_{v_- \in \mathcal{V}_-} \exp\left(\mathbf{v}^1 \cdot \mathbf{v}_-/\tau\right)}. \tag{2}$$

Here $\mathbf{v}^1 = f_\theta(G_1)[v]$ and $\mathbf{v}^2 = f_\theta(G_2)[v]$ are projected representations of node $v$ from two augmented views. $\mathcal{V}_-$ is the set of negative samples of $v$ from the inter- or intra-view [39]. Inspired by BYOL [40], non-contrastive BGRL [5, 41] predicts the target augmented view of the nodes: $\mathcal{L}_V = -1/|\mathcal{V}| \sum_{v \in \mathcal{V}} \mathbf{z}^1 \cdot \mathbf{v}^2/\|\mathbf{z}^1\|\|\mathbf{v}^2\|$, where $\mathbf{z}^1 = g_\phi(\mathbf{v}^1)$ is the prediction of the representation from the target augmented view. An illustration for this scheme is presented in Figure 3 (b).

# 4  Simple Asymmetric Contrastive Learning of Graphs

In this section, we elaborate on our GraphACL. The key idea behind GraphACL is encouraging the encoder to learn representations by simultaneously capturing one-hop neighborhood context and two-hop monophily, which generalizes the homophily assumption for modeling both homophilic and heterophilic graphs. Specifically, GraphACL introduces an additional predictor $g_\phi$ which maps the graph $G$ to node representations that can predict the one-hop neighborhood context from the representation of the central node as shown in Figure 3 (c). Importantly, unlike previous contrastive schemes, which force neighboring nodes to have similar representations, GraphACL directly predicts the original neighborhood signal of each node through an asymmetric design induced by a predictor. This simple asymmetric design allows neighboring nodes to have different representations but still can capture the one-hop neighborhood context and two-hop monophily as illustrated in Figure 2.

## 4.1  Graph Asymmetric Contrastive Learning

Based on the above motivation, for each node $v$, we first propose to learn its representation by capturing its one-hop neighborhood signal. A natural idea of capturing the neighborhood signal is learning the representations of $v$ that can well predict the original features of $v$'s neighbors, i.e., $\mathbf{x}_u$ of neighbor $u$. However, the original features are typically noisy and high-dimensional [42]. To solve this issue, we cast the prediction problem in the representation space, i.e., the representation of the central node should be predictive of representations of its neighbors. Specifically, we adopt the following simple prediction loss induced by an asymmetric predictor on neighbors:

$$\mathcal{L}_\text{PRE} = \frac{1}{|\mathcal{V}|} \sum\nolimits_{v \in \mathcal{V}} \frac{1}{|\mathcal{N}(v)|} \sum\nolimits_{u \in \mathcal{N}(v)} \|g_\phi(\mathbf{v}) - \mathbf{u}\|_2^2, \tag{3}$$

where $\mathbf{v} = f_\theta(G)[v]$ and $\mathbf{u} = f_\theta(G)[u]$ are representations, and $g_\phi(\cdot)$ is a introduced predictor which maps the latent representation $\mathbf{v}$ to reconstruct the neighbor's representation $\mathbf{u}$. Intuitively, in this case, each node is treated as a specific neighbor "context", and nodes with similar distributions over the neighbor "context" are assumed to be similar. Here, we utilize only one single simple $g_\phi$ to predict representations of all $v$'s neighbors and empirically find that it works very well.

This simple prediction objective in Equation (3) can preserve local neighborhood distribution in representations without homophily assumption since we do not directly enforce $\mathbf{v}$ and $\mathbf{u}$ to be similar to each other. To prevent $g_\phi$ from degenerating to the identity function, we consider the representations of central node $v$ and the one-hop neighbor $u$ come from two decoupled encoders: $\mathbf{v} = f_\theta(G)[v]$ and $\mathbf{u} = f_\xi(G)[u]$, where $f_\theta$ is the online identity encoder and $f_\xi$ is the preference target encoder. Importantly, the gradient of the loss in Equation (5) is only used to update the online encoder $f_\theta$, while being blocked in the target encoder $f_\xi$ as shown in Figure 3 (c). The weights of the target network $\xi$ are updated using the exponential moving average of online network weights $\theta$: $\xi \leftarrow \lambda\xi + (1 - \lambda)\theta$ where $\lambda \in [0, 1]$ is the target decay rate. An intuitive illustration for using two encoders is that each node typically plays two roles: the node itself and a specific context of other nodes, corresponding to separated notions of node identity and node preference in real-world graphs. The key principle of decoupled encoders is to allow the identity representation of a node to be more possibly different from the identity representation with which the node prefers to link.

The simple objective in Equation (3) can capture the one-hop neighborhood context without relying on the homophily assumption or requiring graph augmentation as shown in Figure 2 (c). This matches our first motivation in the introduction part. Another remaining problem is why optimizing this simple objective helps capturing 2-hop node similarity, i.e., monophily. Consider a pair of 2-hop neighbors $v$ and $u_2$ which both neighbor on the same node $u$. Intuitively, by enforcing $\mathbf{v}$ and $\mathbf{u}_2$ to reconstruct the same context representation of neighborhood $\mathbf{u}$, we implicitly make their representations similar. Thus, the 2-hop neighbors serve as positive pairs that will be implicitly aligned as shown in Figure 2 (d). We formally prove this intuition in Theorem 2 in our following theoretical analysis.

We note that some contrastive objectives based on augmentations such as BGRL [5] and BYOL [40] also utilize a predictor as shown in Figure 2 (b). However, the idea behind our loss in Equation (3) differs significantly from them. Specifically, they rely on data augmentations, and are built based on the idea that the augmentation can preserve the semantic nature of samples, i.e., the augmented samples have consistent semantic labels with the original ones. Nevertheless, unlike images, it is theoretically difficult to design graph augmentations without changing the semantic nature of nodes [7] and augmentations tend to capture homophily [20, 18]. In contrast, the simple loss in Equation (3) does not rely on augmentations but directly considers modeling one-hop neighborhood distribution. We theoretically prove that our GraphACL maximizes the mutual information between representations and one-hop neighborhood context, and captures two-hop monophily in § 5.

Although this simple neighborhood prediction objective can capture both one-hop neighborhood pattern and two-hop monophily, it may result in a collapsed and trivial encoder: all node representations degenerate to a the same single vector on the hypersphere. The main reason is that the prediction loss operates in a fully flexible latent space, and it can be minimized when the encoder produces a constant representation for all nodes. To address this issue, we introduce an explicit uniformity regularization to further enhance the representation diversity. Specifically, we add the following explicit regularization on representation uniformity into Equation (3):

$$\mathcal{L}_{\text{UNI}} = -\frac{1}{|\mathcal{V}|}\frac{1}{|\mathcal{V}|}\sum_{v \in \mathcal{V}}\sum_{v_- \in \mathcal{V}}\|\mathbf{v} - \mathbf{v}_-\|_2^2, \tag{4}$$

where $\mathbf{v} = f_\theta(G)[v]$ and $\mathbf{v}_- = f_\theta(G)[v_-]$. Here, we consider the representations of negative samples $\mathbf{v}_-$ coming from the same online encoder as central sample. This uniformity loss is typically approximated by randomly sampling $K$ negative samples $v_-$. Intuitively, minimizing this term will push all node representations away from each other and alleviate the representation collapse issue.

**Graph Asymmetric Contrastive Loss**. Directly combining Equations (3) and (4) arrives at a loss function: $\mathcal{L}_{\text{COM}} = \mathcal{L}_{\text{PRE}} + \mathcal{L}_{\text{UNI}}$. However, minimizing this combination loss is an ill-posed problem, it approaches $-\infty$ as we can simply scale the norm of representations to reduce the loss. Even if the representations are normalized, minimizing $\mathcal{L}_{\text{UNI}}$ still leads to a relatively poor performance as shown in our experiments. To address this issue, we instead minimize an upper bound of $\mathcal{L}_{\text{COM}}$, which results in the following simple objective of GraphACL (see Appendix A.1 for details):

$$\mathcal{L}_{\text{A}} = -\frac{1}{|\mathcal{V}|}\sum_{v \in \mathcal{V}}\frac{1}{|\mathcal{N}(v)|}\sum_{u \in \mathcal{N}(v)}\log\frac{\exp(\mathbf{p}^\top\mathbf{u}/\tau)}{\exp(\mathbf{p}^\top\mathbf{u}/\tau) + \sum_{v_- \in \mathcal{V}}\exp(\mathbf{v}^\top\mathbf{v}_-/\tau)}. \tag{5}$$

Here prediction $\mathbf{p} = g_\phi(\mathbf{v})$ with $\mathbf{v} = f_\theta(G)[v]$, representations $\mathbf{u} = f_\xi(G)[u]$ and $\mathbf{v}_- = f_\theta(G)[v_-]$. $\mathcal{L}_A$ is a simple generalization of the graph contrastive loss with representation smoothing in Equation (1) from symmetric view to an asymmetric view. Note that when the predictor $g_\phi$ becomes the

identity function, $\mathcal{L}_A$ degenerates to the GCL loss $\mathcal{L}_S$. We demonstrate in extensive experiments that such asymmetric framework via a simple predictor helps achieve better downstream performance on both homophilic and heterophilic graphs than many GCL methods with prefabricated augmentations.

## 5 Theoretical Analysis

In this section, we provide theoretical understandings of GraphACL. We show that our simple GraphACL can simultaneously capture one-hop neighborhood context and two-hop monophily, which are important for heterophilic graphs. We also establish theoretical guarantees for the downstream performance of the learned representations. *All proofs can be found in Appendix B.*

**Notations.** We denote the random variable of node representations as $V$, and define the mean representations from the one-hop neighborhoods of node $v$ as $\mathbf{z} = \frac{1}{\mathcal{N}(v)} \sum_{u \in \mathcal{N}(v)} \mathbf{u}$. This means that the representation $\mathbf{z}$ profiles the one-hop neighborhood pattern of node $v$. Since two nodes of the same semantic class tend to share similar neighborhood patterns in real-worlds, $\mathbf{z}$ can be viewed sampled from $Z|Y \sim \mathcal{N}(\mathbf{z}_Y, I)$ where $Y$ is the latent semantic class indicating the one-hop pattern of $v$. Given the above, we first demonstrate the rationality of GraphACL in capturing one-hop patterns:

**Theorem 1.** *Minimizing GraphACL's objective in Equation* (5) *with exponential moving average is equivalent to maximizing mutual information between representation $V$ and the one-hop pattern $Y$:*

$$\mathcal{L}_A \geq H(V|Y) - H(V) = -I(V; Y). \tag{6}$$

Theorem 1 indicates that minimizing GraphACL loss in Equation (5) promotes maximizing the mutual information $I(V; Y)$ between representations and one-hop neighborhood context. Next, we theoretically verify that our GraphACL can capture intuition about the two-hop monophily similarity.

**Theorem 2.** *Let $\mathcal{N}_2(v)$ denote the set of two-hop neighbors of $v$. Minimizing the GraphACL objective in Equation* (5) *is approximately minimizing the following alignment loss between two-hop neighbors:*

$$\mathcal{L}_{two\text{-}hop} = \frac{1}{|\mathcal{V}|} \frac{1}{2L} \sum_{v \in \mathcal{V}} \frac{1}{|\mathcal{N}_2(v)|} \sum_{u_2 \in \mathcal{N}_2(v)} \|\mathbf{v} - \mathbf{u}_2\|_2^2, \tag{7}$$

*where $L$ is the L-bi-Lipschitz constant of $g_\phi$: $\forall(\mathbf{v}_1, \mathbf{v}_2)$, $1/L \|\mathbf{v}_1 - \mathbf{v}_2\|_2^2 \leq \|g_\phi(\mathbf{v}_1) - g_\phi(\mathbf{v}_2)\|^2$.*

The above theorem shows that minimizing the GraphACL objective implies a small alignment loss of two-hop neighbors. Thus, GraphACL not only captures the one-hop neighborhood pattern, but also the two-hop monophily implicitly, i.e., encouraging a large similarity between two-hop neighbors.

Next, we theoretically show that the learned representations can achieve better downstream performance. We evaluate the representation by its performance on a multi-class classification task using the mean classifier [43, 44, 45]. Specifically, we consider the classifier $p_W(v) = \arg\max \mathbf{Wv}$, where $\mathbf{v} = f_\theta(G)[v]$ and $\mathbf{W} \in \mathbb{R}^{C \times D}$ is the weight of the linear classification head. The $y_{\text{th}}$ row of $\mathbf{W}$ is the mean $\boldsymbol{\mu}_y$ of representations of nodes with the label $y$: $\boldsymbol{\mu}_y = \mathbb{E}_{v|y}[\mathbf{v}]$. For simplicity, we assume that the node classes are balanced but our results can be easily extended to unbalanced settings by considering a label shift term as shown in the domain adaptation literature [46].

**Theorem 3.** *Let $\hat{h}_2 = \mathcal{H}(\mathcal{G}_2)$ be the homophily ratio of the two-hop graph $\mathcal{G}_2$. Suppose that the downstream task is the M-categorical linear classification and the class is balanced. Then, $\forall q \in$ the hypothesis class with $q = g \circ f$, the upper bound for the classification error on the optimal $\mathbf{v}^*$ is :*

$$P(y_v \neq p_W(\mathbf{v}^*)) \leq 4M^2(4L\mathcal{L}_A(q) + (1 - \hat{h}_2)) + const. \tag{8}$$

This theorem says that the downstream error on representations is bounded by GraphACL loss and homophily ratio $\hat{h}_2$ of the two-hop graph $\mathcal{G}_2$, i.e, monophily. Specifically, a larger monophily ratio $\hat{h}_2$ would provably imply a smaller downstream classification error. Prior work [15] and the statistics of real-world graphs in Appendix C show that although the one-hop graph may be heterophily-dominant, the two-hop graph will always be homophily-dominant ($\hat{h}_2$ is large). This theorem reveals why the simple GraphACL enjoys good performance on both homophilic and heterophilic graphs.

## 6 Experiments

### 6.1 Experimental Settings

**Datasets and Splits.** We conduct experiments on both homophilic and heterophilic graphs. For heterophilic graphs, we adopt Wisconsin, Cornell, Texas [33], Actor, Squirrel, Crocodile, and

Table 1: Node classification accuracy (%) on heterophilic and homophilic graphs. The best and second best performance under each dataset are marked with **boldface** and underline, respectively.

| Method | LINE | VGAE | DGI | GCA | CCA-SSG | BGRL | L-GCL | HGRL | DSSL | SP-GCL | GraphACL |
|---|---|---|---|---|---|---|---|---|---|---|---|
| Squirrel | 38.92±1.58 | 29.13±1.16 | 26.44±1.12 | 48.09±0.21 | 46.76±0.36 | 36.22±1.97 | 52.94±0.88 | 48.31±0.65 | 40.51±0.38 | 52.10±0.67 | **54.05±0.13** |
| Chameleon | 48.59±1.17 | 42.65±1.27 | 60.27±0.70 | 63.66±0.32 | 62.41±0.22 | 64.86±0.63 | 68.74±0.49 | 65.82±0.61 | 66.15±0.32 | 65.28±0.53 | **69.12±0.24** |
| Crocodile | 42.21±1.12 | 45.72±1.53 | 51.25±0.51 | 60.73±0.28 | 56.77±0.39 | 53.87±0.65 | 60.18±0.43 | 61.87±0.45 | 62.98±0.51 | 61.72±0.21 | **66.17±0.24** |
| Actor | 27.55±0.32 | 26.99±1.56 | 28.30±0.76 | 28.77±0.29 | 27.82±0.60 | 28.80±0.54 | 32.55±1.18 | 27.95±0.30 | 28.15±0.31 | 28.94±0.69 | **30.03±0.13** |
| Wisconsin | 37.45±2.51 | 55.67±1.37 | 55.21±1.02 | 59.55±0.81 | 58.46±0.96 | 51.23±1.17 | 65.28±0.52 | 63.90±0.58 | 62.25±0.55 | 60.12±0.39 | **69.22±0.40** |
| Cornell | 43.68±2.17 | 48.73±4.19 | 45.33±6.11 | 52.31±1.09 | 52.17±1.04 | 50.33±2.29 | 52.11±2.37 | 51.78±1.03 | 53.15±1.28 | 52.29±1.21 | **59.33±1.48** |
| Texas | 48.69±1.39 | 50.27±2.21 | 58.53±2.98 | 52.92±0.46 | 59.89±0.78 | 52.77±1.98 | 60.68±1.18 | 61.83±0.71 | 62.11±1.53 | 59.81±1.33 | **71.08±0.34** |
| Roman | 55.42±0.87 | 50.89±0.96 | 63.71±0.63 | 65.79±0.75 | 67.35±0.61 | 68.66±0.39 | 69.74±0.53 | 71.84±0.41 | 71.70±0.54 | 70.88±0.35 | **74.91±0.28** |
| Arxiv-year | 33.21±0.13 | 35.11±0.25 | 39.26±0.72 | 42.96±0.39 | 37.38±0.41 | 43.02±0.62 | 43.92±0.52 | 43.71±0.54 | 45.80±0.57 | 44.11±0.35 | **47.21±0.39** |
| Cora | 68.25±0.31 | 76.30±0.21 | 82.30±0.60 | 82.93±0.42 | 84.00±0.40 | 82.70±0.60 | 84.00±0.35 | 82.52±0.31 | 83.51±0.42 | 83.16±0.13 | **84.20±0.31** |
| Citeseer | 43.92±0.51 | 66.80±0.23 | 71.80±0.70 | 72.19±0.31 | 73.10±0.30 | 71.10±0.80 | 73.26±0.50 | 71.05±0.49 | 73.20±0.51 | 71.96±0.42 | **73.63±0.22** |
| Pubmed | 66.29±0.60 | 75.80±0.40 | 76.80±0.60 | 80.79±0.45 | 81.00±0.40 | 79.60±0.50 | 81.82±0.50 | 79.83±0.31 | 81.25±0.31 | 79.16±0.73 | **82.02±0.15** |
| Computer | 86.50±0.21 | 85.80±0.31 | 83.95±0.47 | 87.85±0.31 | 88.74±0.28 | 89.69±0.37 | 88.72±0.42 | 88.53±0.18 | 89.24±0.23 | 89.68±0.19 | **89.80±0.25** |
| Photo | 89.82±0.20 | 91.50±0.20 | 91.61±0.22 | 91.70±0.10 | 93.14±0.14 | 92.90±0.30 | 93.15±0.47 | 92.85±0.38 | 93.10±0.32 | 92.49±0.31 | **93.31±0.19** |
| Arxiv | 65.22±0.30 | 66.40±0.20 | 70.32±0.25 | 69.37±0.20 | 71.21±0.20 | 71.64±0.24 | 71.33±0.20 | 68.55±0.38 | 69.87±0.47 | 68.25±0.22 | **71.72±0.26** |

Chameleon [33, 47]. We also use two large heterophilic graphs proposed recently: Roman-empire (Roman) [48] and arXiv-year [16] (~170k) nodes. For homophilic graphs, we adopt three citation graphs: Cora, Citeseer and Pubmed [49, 50], and two co-purchase graphs: Computer and Photo [5, 51]. We further include a large-scale homophilic graph Ogbn-Arxiv (Arxiv) [52]. For all datasets, we use the public and standard splits used by cited papers. Detailed descriptions, splits, and one-hop homophily and two-hop monophily statistics of datasets are given in the Appendix C.1.

**Baselines.** We compare GraphACL with a traditional network embedding method: LINE [14], and the recent state-of-the-art self-supervised learning methods: VGAE [9], DGI [2], GCA [4], Local (L)-GCL [12], HGRL [29], BGRL [5], CCA-SSG [6], SP-GCL [53], and DSSL [18]. The descriptions and implementation details of baselines are given in Appendix C.2.

**Evaluation Protocol.** We utilize node and graph classification and node clustering to evaluate the quality of the representation. For classification, we follow the standard linear evaluation protocol [2]. We train a linear classifier on top of the frozen representation and report the test accuracy. For node clustering, we perform k-means clustering on the obtained representations and set the number of clusters to the number of ground truth classes and report the normalized mutual information [54].

**Setup.** For all methods, we use a standard GCN model [49] as the encoder. We randomly initialize the model parameters and use the Adam optimizer to train the encoder. We run experiments with ten random seeds and report the average performance and standard deviation. For fair comparison, for all methods, we select the best configuration of hyperparameters only based on accuracy on the validation set. For baselines that did not report results in part of datasets or do not use standard public data splits [18, 29], we reproduce the results using the official code of the authors. More details on the implementation and the hyperparameter search space can be found in Appendix C.3.

### 6.2 Overall Performance Comparison

Table 1 reports the average node classification accuracy on both heterophilic and homophilic graphs. We provide the graph classification and node clustering results in Appenxi C.6 and C.5, respectively, showing that GraphACL can also effectively adapt to various downstream tasks. Surprisingly, among all methods, our simple GraphACL achieves the best performance in 14 of 15 data sets with various homophily ratios, as shown in Table 1. Specifically, GraphACL achieves significant improvements on most of the heterophilic datasets and comparable performance on homophilic graphs compared with the second-best method. Specifically, GraphACL achieves 4.3% (Roman), 11.6% (Cornell), 14.4% (Texas), 5.1% (Crocodile), and 3.1% (Arxiv-year) relative improvement over the second-best method. We can observe that contrastive strategies (CCA-SSG, GCA, and BGRL) with augmentations can not work well on heterophilic graphs compared to homophilic graphs. This verifies that they still implicitly leverage homophily. In contrast, GraphACL does not need augmentations, and we attribute our significant improvement to modeling the one-hop neighborhood context and two-hop monophily.

### 6.3 Ablation Study and Sensitivity Analysis

**Ablation Study.** We conduct an ablation study in Table 2 to validate our motivation and design, with the following three ablations: **(i)** directly minimize the combination loss $\mathcal{L}_{\text{COM}}$, **(ii)** Removing the

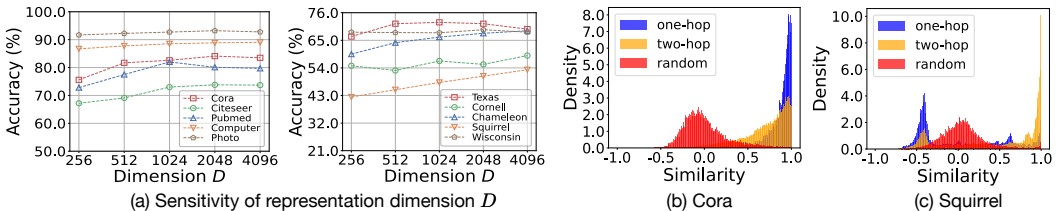

Figure 4: The effect of representation dimension, and the pair-wise similarities of randomly sampled node pairs, one-hop and two-hop neighbors. More results are provided in Appendix C.10.

asymmetric encoder architecture, **(iii)** Removing the uniformity loss with negative samples. We also test the combination **(ii)** & **(iii)** by removing both asymmetric architecture and uniformity loss. First, minimizing the ill-posed combination loss is a valid baseline, but can not achieve better performance and is unstable with large standard deviations. We also find that the model without uniformity loss (ablation **(ii)**) does not achieve the best in homophily graphs, although it does still serve as a strong ablation. When each component is individually applied, the asymmetric architecture alone achieves the very best performance in Cornell and arXiv-year. This confirms our motivation that the asymmetric architecture is of more importance for modeling neighbors in heterophilic graphs where connected nodes have different classes. The full model (last row) achieves the best performance, demonstrating that our designed components are complementary to each other.

**Representation Dimension.** In Figure 4 (a), we analyze the effect of the dimension of node representations. We can observe that having a large dimension can generally lead to better performance for both homophilic and heterophilic graphs. Moreover, we find that the dimension effect is more obvious in heterophilic graphs

Table 2: Ablation studies on the node classification task.

| Baseline | Cora | Pubmed | Cornell | arXiv-year |
|---|---|---|---|---|
| **(i)** minimize $\mathcal{L}_{COM}$ | 83.08±0.75 | 81.31±0.92 | 58.10±2.89 | 45.79±0.62 |
| **(ii)** w/o asymmetric encoder | 82.21±0.60 | 81.05±0.75 | 57.13±0.13 | 45.33±0.22 |
| **(iii)** w/o uniformity loss | 81.39±0.20 | 80.56±0.35 | 57.87±0.27 | 46.54±0.28 |
| **(ii)** & **(iii)** w/o both | 80.85±0.65 | 77.05±0.15 | 42.03±1.21 | 42.71±0.20 |
| **GraphACL** | **84.20±0.31** | **82.02±0.15** | **59.33±1.48** | **47.21±0.39** |

compared to it in homophily graphs. This could be justified by our Theorems 3, which give the lower bound and upper bound of GraphACL loss in terms of the homophily ratio. Specifically, since the two-hop homophily ratio is still smaller in heterophilic graphs compared to homophilic graphs, and a small downstream error requires a small loss. Thus, a large dimension can effectively reduce the lower bound of the training loss and benefit the learning on heterophilic graphs. Training with extremely large dimensions for some graphs may lead to a slight drop of performance as GraphACL may suffer from the over-fitting issue, limiting its generalization performance.

**Other Hyper-parameters.** Figure 5 shows the performance with various decay rate $\lambda$. We observe that (i) $\lambda$ plays an important role in GraphACL. Having a large $\lambda$ typically improves the model performance; (ii) Instantaneously updating the target network, i.e., when $\lambda = 0$, destabilizes training and leads to poor performance. Thus, there is a trade-off between updating the target too often and updating it too slowly. The temperature $\tau$ and the number of negative samples $K$ are not included here since they are not directly relevant to our key motivation and have been employed and evaluated in the recent works [55, 56]. Thus, we provide corresponding results in Appendices C.8 and C.9.

### 6.4 Qualitative Analysis and Case Study

**Similarity Visualization.** Figures 4 (b) and (c) plot the distribution of pair-wise cosine similarities of randomly sampled nodes, one-hop neighbor, and two-hop neighbor pairs based on learned representations. We can observe that, for the homophilic graph, that is, Cora, nodes are forced to have representations similar to those of their neighbors. GraphACL enlarges the similarities between neighbor nodes compared to randomly sampled node pairs, demonstrating that GraphACL can well preserve one-hop neighborhood contexts. Compared to the homophilic graph Cora, GraphACL seeks to further pull the two-hop neighbor nodes together on the heterophilic graph, i.e., Squirrel. This observation matches our motivation and analysis that GraphACL can effectively discover the underlying two-hop monophily, which benefits the learning on heterophilic graphs. Figures 6 (a) and (b) show the cosine similarity between the $\mathbf{v}$ and prediction $g_\phi(\mathbf{v})$ for each node. We find that the cosine similarities are typically smaller than 1, showing that the predictor $g_\phi$ is not an identity matrix after convergence. Moreover, we can observe that, compared to the homophilic graph Cora, the

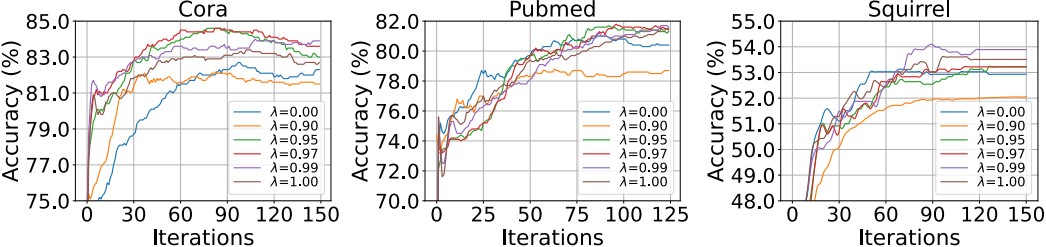

Figure 5: The testing performance varying decay rate $\lambda$. More results are given in Appendix C.7.

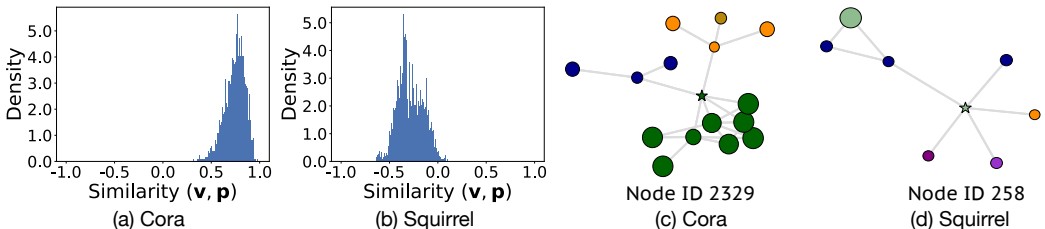

Figure 6: (a) and (b) show the similarity between representation $\mathbf{v}$ and prediction $\mathbf{p}$. (c) and (d) show case studies, where we randomly pick a node with the drastic neighborhood variations and visualize its neighborhood. Node colors denote ground-truth labels. The size of the node is proportional to its representation similarity to the central node denoted as star. See Appendix C.11 for more results.

similarities on the heterophilic graph Squirrel are smaller. This verifies that GraphACL automatically differentiates node identity and context representations, which is important for heterophilic graphs.

**Case Study.** In Figures 6 (c) and (d), we randomly sample the two-hop subgraph of a central node and calculate the cosine similarity based on learned representations between the central node and neighbor nodes. We can observe that GraphACL can successfully identify the node whose local neighborhood patterns are similar to the central node on both homophilic and heterophilic graphs. Additionally, our GraphACL can pull two-hop neighbor nodes, which share a similar semantic class to the central nodes, instead of favoring nearby one-hop neighbor nodes. These observations endorse our intuition that heterophilic graphs can not benefit much from one-hop neighbor nodes, and our GraphACL can effectively capture the latent semantic information related to two-hop monophily.

# 7   Conclusions

We propose a simple contrastive learning framework named GraphACL for homophilic and heterophilic graphs. The key idea of GraphACL is to capture both a local neighborhood context of one hop and a monophily similarity of two hops in one single objective. Specifically, we present a simple asymmetric contrastive objective for node representation learning. We provide a theoretical understanding of GraphACL, which shows that GraphACL can explicitly maximize mutual information between representations and one-hop neighborhood patterns. We show that GraphACL also implicitly aligns the two-hop neighbors and enjoys a good downstream performance. Extensive experiments on 15 graph benchmarks show that GraphACL significantly outperforms state-of-the-art methods. We hope that the straightforward nature of our approach serves as a reminder to the community to reevaluate simpler alternatives that may have been overlooked, thus inspiring future research.

# Acknowledgments

This work supported by, or in part by, the National Science Foundation (NSF) under grant number IIS-1909702, the Army Research Office (ARO) under grant number W911NF21-1-0198, Department of Homeland Security (DNS) CINA under grant number E205949D, and Cisco Faculty Research Award. The findings in this paper do not necessarily reflect the view of the funding agencies.

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

# A  The Omitted Details of GraphACL

## A.1  The Loss of Graph Asymmetric Contrastive Learning

We provide details of the derivations of GraphACL in Equation (5) and show that minimizing GraphACL is approximately to minimize the combination of prediction and uniformity losses.

$$
\begin{aligned}
\mathcal{L}_\text{A} &= -\frac{1}{|\mathcal{V}|} \sum_{v \in \mathcal{V}} \frac{1}{|\mathcal{N}(v)|} \sum_{u \in \mathcal{N}(v)} \log \frac{\exp(\mathbf{p}^\top \mathbf{u}/\tau)}{\exp\left(\mathbf{p}^\top \mathbf{u}/\tau\right) + \sum_{v_- \in \mathcal{V}} \exp\left(\mathbf{v}^\top \mathbf{v}_-/\tau\right)} \\
&= \frac{1}{|\mathcal{V}|} \sum_{v \in \mathcal{V}} \frac{1}{|\mathcal{N}(v)|} \sum_{u \in \mathcal{N}(v)} -\frac{\mathbf{p}^\top \mathbf{u}}{\tau} + \log\left(\exp\left(\mathbf{p}^\top \mathbf{u}/\tau\right) + \sum_{v_- \in \mathcal{V}} \exp\left(\mathbf{v}^\top \mathbf{v}_-/\tau\right)\right) \\
&\geq \frac{1}{|\mathcal{V}|} \sum_{v \in \mathcal{V}} \frac{1}{|\mathcal{N}(v)|} \sum_{u \in \mathcal{N}(v)} -\frac{\mathbf{p}^\top \mathbf{u}}{\tau} + \log \sum_{v_- \in \mathcal{V}} \exp\left(\mathbf{v}^\top \mathbf{v}_-/\tau\right) && (9) \\
&= \frac{1}{|\mathcal{V}|} \sum_{v \in \mathcal{V}} \frac{1}{|\mathcal{N}(v)|} \sum_{u \in \mathcal{N}(v)} -\frac{\mathbf{p}^\top \mathbf{u}}{\tau} + \log \sum_{v_- \in \mathcal{V}} \exp\left(\mathbf{v}^\top \mathbf{v}_-/\tau\right) \\
&= \frac{1}{|\mathcal{V}|} \sum_{v \in \mathcal{V}} \frac{1}{|\mathcal{N}(v)|} \sum_{u \in \mathcal{N}(v)} -\frac{\mathbf{p}^\top \mathbf{u}}{\tau} + \log \sum_{v_- \in \mathcal{V}} \frac{\exp\left(\mathbf{v}^\top \mathbf{v}_-/\tau\right)}{|\mathcal{V}|} \cdot |\mathcal{V}| \\
&\geq \frac{1}{|\mathcal{V}|} \sum_{v \in \mathcal{V}} \frac{1}{|\mathcal{N}(v)|} \sum_{u \in \mathcal{N}(v)} -\frac{\mathbf{p}^\top \mathbf{u}}{\tau} + \log |\mathcal{V}| + \sum_{v_- \in \mathcal{V}} \log \frac{\exp\left(\mathbf{v}^\top \mathbf{v}_-/\tau\right)}{|\mathcal{V}|} && (10) \\
&\overset{\text{c}}{=} \frac{1}{|\mathcal{V}|} \sum_{v \in \mathcal{V}} \frac{1}{|\mathcal{N}(v)|} \sum_{u \in \mathcal{N}(v)} -\frac{\mathbf{p}^\top \mathbf{u}}{\tau} + \frac{1}{|\mathcal{V}|} \sum_{v_- \in \mathcal{V}} \log \exp\left(\mathbf{v}^\top \mathbf{v}_-/\tau\right) \\
&\overset{\text{c}}{=} \frac{1}{|\mathcal{V}|} \sum_{v \in \mathcal{V}} \frac{1}{|\mathcal{N}(v)|} \sum_{u \in \mathcal{N}(v)} -\mathbf{p}^\top \mathbf{u} + \frac{1}{|\mathcal{V}|} \sum_{v_- \in \mathcal{V}} \mathbf{v}^\top \mathbf{v}_- \\
&\overset{\text{c}}{=} \frac{1}{|\mathcal{V}|} \sum_{v \in \mathcal{V}} \frac{1}{|\mathcal{N}(v)|} \sum_{u \in \mathcal{N}(v)} \|\mathbf{p} - \mathbf{u}\|_2^2 - \frac{1}{|\mathcal{V}|} \frac{1}{|\mathcal{V}|} \sum_{v \in \mathcal{V}} \sum_{v_- \in \mathcal{V}} \|\mathbf{v} - \mathbf{v}_-\|_2^2, && (11)
\end{aligned}
$$

where the symbol $\overset{\text{c}}{=}$ indicates equality up to a multiplicative and/or additive constant. Here, we utilize Jensen's inequality in Equation (10). Equation (11) holds because $\mathbf{p}$ and $\mathbf{u}$ are both $\ell_2$-normalized. Given the above, we can conclude that minimizing GraphACL objective is approximately to minimizing the combination of the prediction and uniformity losses.

# B  Proofs in Section 5

## B.1  Proof of Theorem 1

**Theorem 1.** *Minimizing GraphACL's objective in Equation (5) with exponential moving average is equivalent to maximizing mutual information between representation $V$ and the one-hop pattern $Y$:*

$$
\mathcal{L}_A \geq H(V|Y) - H(V) = -I(V;Y). \tag{12}
$$

*Proof.* According to the derivations in Appendix A.1, we have:

$$
\begin{aligned}
\mathcal{L}_\text{A} &\geq \frac{1}{|\mathcal{V}|} \sum_{v \in \mathcal{V}} \frac{1}{|\mathcal{N}(v)|} \sum_{u \in \mathcal{N}(v)} -\mathbf{p}^\top \mathbf{u} + \frac{1}{|\mathcal{V}|} \sum_{v_- \in \mathcal{V}} \mathbf{v}^\top \mathbf{v}_- \\
&\overset{\text{c}}{=} \frac{1}{|\mathcal{V}|} \sum_{v \in \mathcal{V}} \left\|\mathbf{p} - \frac{1}{|\mathcal{N}(v)|} \sum_{u \in \mathcal{N}(v)} \mathbf{u}\right\|_2^2 - \frac{1}{|\mathcal{V}|} \sum_{v \in \mathcal{V}} \sum_{v_- \in \mathcal{V}} \|\mathbf{v} - \mathbf{v}_-\|_2^2 && (13) \\
&= \frac{1}{|\mathcal{V}|} \sum_{v \in \mathcal{V}} \|\mathbf{p} - \mathbf{z}\|_2^2 - \frac{1}{|\mathcal{V}|} \frac{1}{|\mathcal{V}|} \sum_{v \in \mathcal{V}} \sum_{v_- \in \mathcal{V}} \|\mathbf{v} - \mathbf{v}_-\|_2^2, && (14)
\end{aligned}
$$

where $\mathbf{z} = \frac{1}{\mathcal{N}(v)} \sum_{u \in \mathcal{N}(v)} \mathbf{u}$ is the mean representations from one-hop neighborhoods of node $v$. Equation (13) holds because that gradient of $\mathbf{u}$ is being blocked in the target encoder with exponential moving average; thus it can be viewed as a constant that depends only on the target encoder but not the variable $f_\theta$. Since two nodes of the same semantic class tend to share similar one-hop neighborhood patterns even in non-homophilous graphs, thus we can view the neighborhood representations $z$ is

sampled from the conditional distribution given the pseudo label $Y$ which indicates the one-hop neighborhood pattern: $Z|Y \sim \mathcal{N}(\mathbf{z}_Y, I)$. As demonstrated in [57], we can interpret the first term as a conditional cross-entropy between $V$ and another random variable $Z$ whose conditional distribution given the pseudo-label $Y$ indicates the one-hop neighborhood pattern: $Z|Y \sim \mathcal{N}(\mathbf{z}_Y, I)$:

$$\frac{1}{|\mathcal{V}|} \sum_{v \in \mathcal{V}} \|\mathbf{p} - \mathbf{z}\|_2^2 \overset{c}{=} H(V; Z|Y) = H(V|Y) + \mathcal{D}_{KL}(V\|Z|Y) \geq H(V|Y), \quad (15)$$

where $H(\cdot)$ and $\mathcal{D}_{KL}(\cdot)$ denote the entropy and KL-divergence, respectively. Minimizing the first term in Equation (14) is approximately minimizing $H(V|Y)$. We then inspect the second term in Equation (14). As shown in [58, 57], the second term is close to the differential entropy estimator:

$$\frac{1}{|\mathcal{V}|} \frac{1}{|\mathcal{V}|} \sum_{v \in \mathcal{V}} \sum_{v_- \in \mathcal{V}} \|\mathbf{v} - \mathbf{v}_-\|_2^2 \propto H(V). \quad (16)$$

As a result, minimizing the loss of GraphACL can be seen as a proxy for maximizing the mutual information between the representations $V$ and the pseudo-labels $Y$ which indicates the neighborhood patterns of one hop nodes. Thus, the proof is completed. $\square$

## B.2 Proof of Theorem 2

**Theorem 2** *Let $\mathcal{N}_2(v)$ denote the set of two-hop neighbors of $v$. Minimizing the GraphACL objective in Equation* (5) *is approximately minimizing the following alignment loss between two-hop neighbors:*

$$\mathcal{L}_{\text{two-hop}} = \frac{1}{|\mathcal{V}|} \frac{1}{2L} \sum_{v \in \mathcal{V}} \frac{1}{|\mathcal{N}_2(v)|} \sum_{u_2 \in \mathcal{N}_2(v)} \|\mathbf{v} - \mathbf{u}_2\|_2^2, \quad (17)$$

*where $L$ is the L-bi-Lipschitz constant of $g_\phi$: $\forall (\mathbf{v}_1, \mathbf{v}_2)$, $1/L \|\mathbf{v}_1 - \mathbf{v}_2\|_2^2 \leq \|g_\phi(\mathbf{v}_1) - g_\phi(\mathbf{v}_2)\|^2$.*

*Proof.* Given the derivations in Appendix A.1, we have:

$$\mathcal{L}_A \overset{c}{=} \frac{1}{|\mathcal{V}|} \sum_{v \in \mathcal{V}} \frac{1}{|\mathcal{N}(v)|} \sum_{u \in \mathcal{N}(v)} -\frac{\mathbf{p}^\top \mathbf{u}}{\tau} + \log \left( \exp \left( \mathbf{p}^\top \mathbf{u}/\tau \right) + \sum_{v_- \in \mathcal{V}} \exp \left( \mathbf{v}^\top \mathbf{v}_-/\tau \right) \right)$$

$$\geq -\frac{1}{|\mathcal{V}|} \sum_{v \in \mathcal{V}} \frac{1}{|\mathcal{N}(v)|} \sum_{u \in \mathcal{N}(v)} \frac{\mathbf{u}^\top g_\phi(\mathbf{v})}{\tau} \overset{c}{=} -\frac{1}{|\mathcal{V}|} \sum_{v \in \mathcal{V}} \frac{1}{|\mathcal{N}(v)|} \sum_{u \in \mathcal{N}(v)} \mathbf{u}^\top g_\phi(\mathbf{v}), \quad (18)$$

where the last line holds because $v \in \mathcal{V}$: $\sum_{v_- \in \mathcal{V}} \exp \left( \mathbf{v}^\top \mathbf{v}_-/\tau \right) > \exp(\mathbf{v}^\top \mathbf{v}) = e > 1$. To conduct the proof, we first formulate the degree-related predictions and representations as two matrices: $\tilde{\mathbf{P}}$ and $\tilde{\mathbf{U}}$. Here, $\tilde{\mathbf{U}}_{v_1} = \sqrt{d_u}\mathbf{u}$ is the $u$-th row of the matrix $\tilde{\mathbf{U}} \in \mathbb{R}^{N \times D}$ and $\tilde{\mathbf{P}}_v = \sqrt{d_v} \cdot g_\phi(\mathbf{v})$ is the $v$-th row of the matrix $\tilde{\mathbf{P}} \in \mathbb{R}^{N \times D}$. Recall that the normalized adjacency matrix $\tilde{\mathbf{A}} = \mathbf{D}^{-1/2}\mathbf{A}\mathbf{D}^{-1/2}$ where $\mathbf{D}$ is the diagonal degree matrix. With the above and Equation (18), we have:

$$\mathcal{L}_A \geq -\frac{1}{|\mathcal{V}|} \sum_{v \in \mathcal{V}} \frac{1}{|\mathcal{N}(v)|} \sum_{u \in \mathcal{N}(v)} g_\phi(\mathbf{v})^\top \mathbf{u}$$

$$= -\frac{1}{|\mathcal{V}|} \sum_{v \in \mathcal{V}} \frac{1}{|\mathcal{N}(v)|} \sum_{u \in \mathcal{N}(v)} \frac{1}{\sqrt{d_v}\sqrt{d_u}} \sqrt{d_v} g_\phi(\mathbf{v}) \cdot \sqrt{d_u}\mathbf{u}$$

$$\overset{c}{=} -\frac{1}{|\mathcal{V}|} \sum_{v \in \mathcal{V}} \frac{d_m}{|\mathcal{N}(v)|} \sum_{u \in \mathcal{N}(v)} \frac{1}{\sqrt{d_v}\sqrt{d_u}} \sqrt{d_v} g_\phi(\mathbf{v}) \cdot \sqrt{d_u}\mathbf{u}$$

$$\geq -\frac{1}{|\mathcal{V}|} \sum_{v \in \mathcal{V}} \sum_{u \in \mathcal{N}(v)} \frac{1}{\sqrt{d_v}\sqrt{d_u}} \sqrt{d_v} g_\phi(\mathbf{v}) \cdot \sqrt{d_u}\mathbf{u}$$

$$= -\frac{1}{|\mathcal{V}|} \text{tr}(\tilde{\mathbf{A}}\tilde{\mathbf{P}}\tilde{\mathbf{U}}^\top) \geq -\frac{1}{|\mathcal{V}|} \frac{1}{2}(\|\tilde{\mathbf{A}}\tilde{\mathbf{P}}\|_2^2 + \|\tilde{\mathbf{U}}^\top\|_2^2) \quad (19)$$

$$= -\frac{1}{|\mathcal{V}|} \frac{1}{2} \text{tr}(\tilde{\mathbf{P}}\tilde{\mathbf{P}}^\top \tilde{\mathbf{A}}^\top \tilde{\mathbf{A}}) - \frac{1}{|\mathcal{V}|} \frac{1}{2} \sum_{u \in \mathcal{V}} d_u \|\mathbf{u}\|_2^2 \overset{c}{=} -\frac{1}{|\mathcal{V}|} \frac{1}{2} \text{tr}(\tilde{\mathbf{P}}\tilde{\mathbf{P}}^\top \tilde{\mathbf{A}}^\top \tilde{\mathbf{A}}), \quad (20)$$

where $d_m$ is the maximum degree of nodes and $\text{tr}(\cdot)$ denotes the matrix trace. In Equation (19), we utilize the inequality: $\text{tr}(\mathbf{PQ}) \leq \frac{1}{2} \left( \|\mathbf{P}\|_2^2 + \|\mathbf{Q}\|_2^2 \right)$ for any two matrices $\mathbf{P} \in \mathbb{R}^{m \times n}$ and $\mathbf{Q} \in \mathbb{R}^{n \times t}$. Equation (20) holds as $\mathbf{u}$ is normalized. Since the trace is the sum of elements on the main diagonal

of the matrix, the $v$-th diagonal value of $(\tilde{\mathbf{P}}\tilde{\mathbf{P}}^\top\tilde{\mathbf{A}}^\top\tilde{\mathbf{A}})$ is:

$$(\tilde{\mathbf{P}}\tilde{\mathbf{P}}^\top\tilde{\mathbf{A}}^\top\tilde{\mathbf{A}})_{v,v} = \sum\nolimits_{u_2}(\tilde{\mathbf{A}}^\top\tilde{\mathbf{A}})_{v,u_2}(\tilde{\mathbf{P}}\tilde{\mathbf{P}}^\top)_{u_2,v} = \sum_{u_2\in\mathcal{V}}\sum_{u\in\mathcal{N}(v)\cap\mathcal{N}(u_2)}\tilde{\mathbf{A}}^T_{v,u}\tilde{\mathbf{A}}_{u,u_2}\sqrt{d_v}\sqrt{d_{u_2}}\mathbf{p}_v^\top\mathbf{p}_{u_2}$$

$$= \sum\nolimits_{u\in\mathcal{N}(v)}\sum\nolimits_{u_2\in\mathcal{N}(u)}\frac{1}{d_u}\frac{1}{\sqrt{d_v}\sqrt{d_{u_2}}}\sqrt{d_v}\sqrt{d_{u_2}}\mathbf{p}_v^\top\mathbf{p}_{u_2}$$

$$= \sum\nolimits_{u\in\mathcal{N}(v)}\sum\nolimits_{u_2\in\mathcal{N}(u)}\frac{1}{|\mathcal{N}(u)|}\mathbf{p}_v^\top\mathbf{p}_{u_2} \overset{c}{=} -\sum\nolimits_{u\in\mathcal{N}(v)}\sum\nolimits_{u_2\in\mathcal{N}(u)}\frac{1}{|\mathcal{N}(u)|}\|\mathbf{p}_v-\mathbf{p}_{u_2}\|_2^2$$

$$= -\sum\nolimits_{u\in\mathcal{N}(v)}\sum\nolimits_{u_2\in\mathcal{N}(u)}\frac{1}{|\mathcal{N}(u)|}\|g_\phi(\mathbf{v})-g_\phi(\mathbf{u}_2)\|_2^2$$

$$\leq -\frac{1}{L}\sum\nolimits_{u_2\in\mathcal{N}(u)}\sum\nolimits_{u\in\mathcal{N}(v)}\frac{1}{|\mathcal{N}(u)|}\|\mathbf{v}-\mathbf{u}_2\|_2^2, \tag{21}$$

where Equation (21) holds is because the decoder is $L$-bi-Lipschitz: $\forall(\mathbf{x}_1,\mathbf{x}_2),\,1/L\,\|\mathbf{x}_1-\mathbf{x}_2\|^2 \leq \|g_\phi(\mathbf{x}_1)-g_\phi(\mathbf{x}_2)\|^2$. Combining Equations (18), (20) and (21), we have:

$$\min_{\theta,\phi}\mathcal{L}_A \Rightarrow \min_{\theta,\phi}-\frac{1}{|\mathcal{V}|}\sum\nolimits_{v\in\mathcal{V}}\frac{1}{|\mathcal{N}(v)|}\sum\nolimits_{u\in\mathcal{N}(v)}\mathbf{u}^\top g_\phi(\mathbf{v})$$

$$\Rightarrow \min_{\theta,\phi}-\frac{1}{|\mathcal{V}|}\frac{1}{2}\mathrm{tr}(\tilde{\mathbf{P}}\tilde{\mathbf{P}}^\top\tilde{\mathbf{A}}^\top\tilde{\mathbf{A}}) \Leftrightarrow \min_{\theta,\phi}-\frac{1}{|\mathcal{V}|}\frac{1}{2}\sum\nolimits_{v\in\mathcal{V}}(\tilde{\mathbf{P}}\tilde{\mathbf{P}}^\top\tilde{\mathbf{A}}^\top\tilde{\mathbf{A}})_{v,v}$$

$$\Rightarrow \min_{\theta,\phi}\frac{1}{2L}\frac{1}{|\mathcal{V}|}\sum\nolimits_{v\in\mathcal{V}}\sum\nolimits_{u\in\mathcal{N}(v)}\frac{1}{|\mathcal{N}(u)|}\sum\nolimits_{u_2\in\mathcal{N}(u)}\|\mathbf{v}-\mathbf{u}_2\|_2^2$$

$$\Leftrightarrow \min_{\theta,\phi}\frac{1}{2L}\frac{1}{|\mathcal{V}|}\sum\nolimits_{v\in\mathcal{V}}\sum\nolimits_{u\in\mathcal{N}(v)}\frac{d_m}{|\mathcal{N}(u)|}\sum\nolimits_{u_2\in\mathcal{N}(u)}\|\mathbf{v}-\mathbf{u}_2\|_2^2$$

$$\Rightarrow \min_{\theta,\phi}\frac{1}{2L}\frac{1}{|\mathcal{V}|}\sum\nolimits_{v\in\mathcal{V}}\sum\nolimits_{u_2\in\mathcal{N}_2(v)}\|\mathbf{v}-\mathbf{u}_2\|_2^2$$

$$\Rightarrow \min_{\theta,\phi}\frac{1}{2L}\frac{1}{|\mathcal{V}|}\sum\nolimits_{v\in\mathcal{V}}\frac{1}{|\mathcal{N}_2(v)|}\sum\nolimits_{u_2\in\mathcal{N}_2(v)}\|\mathbf{v}-\mathbf{u}_2\|_2^2, \tag{22}$$

where $\min_\theta f_1 \Rightarrow \min_\theta f_2$ represents that minimizing function $f_1$ with respect to $\theta$ is equivalent to maximizing function $f_2$ with respect to $\theta$. Thus, the proof is completed. $\square$

## B.3 Proof of Theorem 3

**Theorem 3** *Let $\hat{h}_2 = \mathcal{H}(\mathcal{G}_2)$ be the homophily ratio of the two-hop graph $\mathcal{G}_2$. Suppose that the downstream task is the $M$-categorical linear classification and the class is balanced. Then, $\forall q \in$ the hypothesis class with $q = g \circ f$, the upper bound for the classification error on the optimal $\mathbf{v}^*$ is:*

$$P\left(y_v \neq p_W(\mathbf{v}^*)\right) \leq 4M^2(4L\mathcal{L}_A(q) + (1-\hat{h}_2)) + const, \tag{23}$$

*Proof.* We define the adjacency matrix of the two-hop graph of $G$ as $\mathbf{A}_2$, i.e., $(\mathbf{A}_2)_{ik} = 1$ if there exists node $j$ such that $e_{ij}\in\mathcal{E}$ and $e_{jk}\in\mathcal{E}$, otherwise $(\mathbf{A}_2)_{ik} = 0$. The normalized form of $\mathbf{A}_2$ is $\hat{\mathbf{A}}_2 = \hat{\mathbf{D}}^{-1/2}\mathbf{A}_2\hat{\mathbf{D}}_2$ where $\hat{\mathbf{D}}_2$ is the diagonal degree matrix of the two-hop graph. $\mathcal{N}_2(v)$ is the set of the neighbors of $v$ on the two-hop graph. We also define the label matrix of all nodes as $\mathbf{Y}\in\mathbb{R}^{N\times M}$ and the representation matrix $\mathbf{V}\in\mathbb{R}^{M\times D}$ with the $v$-th node representation as $\mathbf{V}_v = \mathbf{v}$. Let $c_i$ be the number of nodes belonging to class $i$. For any class $i$, we have $c_i = c = \frac{|\mathcal{V}|}{M}$ since we assume the ideal balanced setting for simplicity. However, our theoretical results can be easily extended to unbalanced settings by explicitly considering the shift of label distributions [46]. Given the above definitions, we have the following:

$$\mathbb{E}_{v,\mathbf{y}_v}\|\mathbf{y}_v-\mathbf{W}\mathbf{v}\|_2^2 \simeq \frac{1}{|\mathcal{V}|}\sum\nolimits_{v\in\mathcal{V}}\|\mathbf{y}_v-\mathbf{W}\mathbf{v}\|_2^2 = \frac{1}{|\mathcal{V}|}\|\mathbf{Y}-\mathbf{V}\mathbf{W}^\top\|_2^2,$$

$$= \frac{1}{|\mathcal{V}|}\|\mathbf{Y}-\mathbf{A}_2\mathbf{C}+\mathbf{A}_2\mathbf{C}-\mathbf{V}\mathbf{W}^\top\|_2^2,$$

$$\leq \frac{1}{|\mathcal{V}|}\left(2\|\mathbf{Y}-\mathbf{A}_2\mathbf{C}\|_2^2+2\|\mathbf{A}_2\mathbf{C}-\mathbf{V}\mathbf{W}^\top\|_2^2\right), \tag{24}$$

where $\mathbf{C}_{v,i} = c_i^{-1}\mathbb{1}_{y_v=i}$ and $\mathbb{1}$ is the indicator function. $y_v$ is the label index of node $v$ and $\mathbf{y}_v$ is the one-hot vector of label $y_v$. The last line holds because $\|\mathbf{P}+\mathbf{Q}\|_2^2 < 2\|\mathbf{P}\|_2^2 + 2\|\mathbf{Q}\|_2^2$ for any

two matrices $\mathbf{P}$ and $\mathbf{Q}$ of the same dimension. The $(v, i)$-th elements of matrices $\mathbf{Y}$ and $\mathbf{A}_2\mathbf{C}$ are: $\mathbf{Y}_{v,i} = \mathbb{1}_{y_v=i}$ and $(\mathbf{A}_2\mathbf{C})_{v,i} = \sum_{u_2 \in \mathcal{N}_2(v)} c_i^{-1} \mathbb{1}_{y_{u_2}=i}$. If $y_v = i$, we have:

$$(\mathbf{Y} - \mathbf{A}_2\mathbf{C})_{v,i} = \mathbb{1}_{y_v=i} - \sum_{u_2 \in \mathcal{N}_2(v)} \frac{1}{c_i} \mathbb{1}_{y_{u_2}=i} = 1 - \sum_{u_2 \in \mathcal{N}_2(v)} \frac{1}{c_i} \mathbb{1}_{y_{u_2}=i}, \qquad (25)$$

Similarly, for $y_v \neq i$, we have:

$$(\mathbf{Y} - \mathbf{A}_2\mathbf{C})_{v,i} = \mathbb{1}_{y_v=i} - \sum_{u_2 \in \mathcal{N}_2(v)} \frac{1}{c_i} \mathbb{1}_{y_{u_2}=i} = -\sum_{u_2 \in \mathcal{N}_2(v)} \frac{1}{c_i} \mathbb{1}_{y_{u_2}=i}. \qquad (26)$$

Given the above, we further have:

$$
\begin{aligned}
\|\mathbf{Y} - \mathbf{A}_2\mathbf{C}\|_2^2 &= \sum_v \|(\mathbf{Y} - \mathbf{A}_2\mathbf{C})_v\|_2^2 = \sum_v \sum_{i=1}^M (\mathbf{Y} - \mathbf{A}_2\mathbf{C})_{v,i}^2 \\
&= \sum_v \sum_{i=1}^M \mathbb{1}_{y_v=i}(1 - \sum_{u_2 \in \mathcal{N}_2(v)} \frac{1}{c_i} \mathbb{1}_{y_{u_2}=i})^2 + (1 - \mathbb{1}_{y_v=i})(-\sum_{v_u \in \mathcal{N}_2(v)} \frac{1}{c_i} \mathbb{1}_{y_{u_2}=i})^2 \\
&= \sum_v (1 - \sum_{u_2 \in \mathcal{N}_2(v)} \frac{1}{c_i} \mathbb{1}_{y_{u_2}=y_v})^2 + \sum_{i=1}^M \mathbb{1}_{y_v \neq i}(\sum_{u_2 \in \mathcal{N}_2(v)} \frac{1}{c_i} \mathbb{1}_{y_{u_2}=i})^2 \\
&\leq \sum_v (1 - \sum_{v_2 \in \mathcal{N}_2(v)} \frac{1}{c_i} \mathbb{1}_{y_{u_2}=y_v})^2 + (\sum_{i=1}^M \mathbb{1}_{y_v \neq i} \sum_{u_2 \in \mathcal{N}_2(v)} \frac{1}{c_i} \mathbb{1}_{y_{u_2}=i})^2 \\
&= \sum_v (1 - \sum_{u_2 \in \mathcal{N}_2(v)} \frac{1}{c_i} \mathbb{1}_{y_{u_2}=y_v})^2 + (\sum_{u_2 \in \mathcal{N}_2(v)} (\sum_{i=1}^M \mathbb{1}_{y_v \neq i} \frac{1}{c_i} \mathbb{1}_{y_{u_2}=i}))^2 \\
&= \sum_v 1 + \frac{MM}{|\mathcal{V}||\mathcal{V}|} (\sum_{u_2 \in \mathcal{N}_2(v)} (\sum_{i=1}^M \mathbb{1}_{y_v \neq i} \mathbb{1}_{y_{u_2}=i}))^2 \\
&= \sum_v 1 + \frac{MM}{|\mathcal{V}||\mathcal{V}|} (\sum_{u_2 \in \mathcal{N}_2(v)} \mathbb{1}_{y_{u_2} \neq y_v})^2 \leq |\mathcal{V}| + \frac{MM}{|\mathcal{V}|} \sum_v \sum_{u_2 \in \mathcal{N}_2(v)} \mathbb{1}_{y_{u_2} \neq y_v}.
\end{aligned}
$$
$$(27)$$

For the second term $\|\mathbf{A}_2\mathbf{C} - \mathbf{V}\mathbf{W}^\top\|_2^2$ in Equation (24), we have:

$$
\begin{aligned}
\|\mathbf{A}_2\mathbf{C} - \mathbf{V}\mathbf{W}^\top\|_2^2 &= \|(\mathbf{A}_2 - \mathbf{V}\mathbf{V}^\top + \mathbf{V}\mathbf{V}^\top)\mathbf{C} - \mathbf{V}\mathbf{W}^\top\|_2^2 && (28) \\
&= \|(\mathbf{A}_2 - \mathbf{V}\mathbf{V}^\top)\mathbf{C} + \mathbf{V}(\mathbf{V}^\top\mathbf{C} - \mathbf{W}^\top)\|_2^2 \\
&\leq 2\|(\mathbf{A}_2 - \mathbf{V}\mathbf{V}^\top)\mathbf{C}\|_2^2 + 2\|\mathbf{V}(\mathbf{V}^\top\mathbf{C} - \mathbf{W}^\top)\|_2^2 \\
&\leq 2\|(\mathbf{A}_2 - \mathbf{V}\mathbf{V}^\top)\|_2^2\|\mathbf{C}\|_2^2 + 2\|\mathbf{V}\|_2^2\|(\mathbf{V}^\top\mathbf{C} - \mathbf{W}^\top)\|_2^2 \\
&\leq 2\|(\mathbf{A}_2 - \mathbf{V}\mathbf{V}^\top)\|_2^2\|\mathbf{C}\|_2^2 + 2\|\mathbf{V}\|_2^2\|(\mathbf{V}^\top\mathbf{C} - \mathbf{W}^\top)\|_2^2 \\
&= 2\|(\mathbf{A}_2 - \mathbf{V}\mathbf{V}^\top)\|_2^2\|\mathbf{C}\|_2^2 && (29) \\
&= 2\frac{MM}{|\mathcal{V}|}\|(\mathbf{A}_2 - \mathbf{V}\mathbf{V}^\top)\|_2^2, && (30)
\end{aligned}
$$

where Equation (29) holds because the $i_\text{th}$ row of $\mathbf{W}$ is the mean of representations of nodes with label $i$: $\|(\mathbf{V}^\top\mathbf{C} - \mathbf{W}^\top)\|_2^2 = 0$. Combining Equations (24), (27), and (30), we have:

$$
\begin{aligned}
\mathbb{E}_{v,\mathbf{y}_v} \|\mathbf{y}_v - \mathbf{W}\mathbf{v}\|_2^2 &\leq \frac{1}{|\mathcal{V}|} \left( 2\|\mathbf{Y} - \mathbf{A}_2\mathbf{C}\|_2^2 + 2\|\mathbf{A}_2\mathbf{C} - \mathbf{V}\mathbf{W}^\top\|_2^2 \right) \\
&\leq 4\frac{MM}{|\mathcal{V}||\mathcal{V}|}\|(\mathbf{A}_2 - \mathbf{V}\mathbf{V}^\top)\|_2^2 + 2(1 + \frac{MM}{|\mathcal{V}||\mathcal{V}|} \sum_v \sum_{v_2 \in \mathcal{N}_2(v)} \mathbb{1}_{y_{v_2} \neq y_v}) \\
&\leq 4\frac{MM}{|\mathcal{V}||\mathcal{V}|}\|(\mathbf{A}_2 - \mathbf{V}\mathbf{V}^\top)\|_2^2 + 2(1 + \frac{MM}{|\mathcal{V}||\mathcal{V}|} \sum_v \frac{|\mathcal{V}|}{|\mathcal{N}_2(v)|} \sum_{v_2 \in \mathcal{N}_2(v)} \mathbb{1}_{y_{v_2} \neq y_v}), \\
&\leq 4\frac{MM}{|\mathcal{V}||\mathcal{V}|}\|(\mathbf{A}_2 - \mathbf{V}\mathbf{V}^\top)\|_2^2 + 2M^2(1 - \hat{h}_2) + 2, && (31)
\end{aligned}
$$

where $\hat{h}_2 = \mathcal{H}(\mathcal{G}_2)$ is the homophily ratio of the two-hop graph. Based on the derivations in Theorem 2 and Equation (9), we have:

$$\mathcal{L}_{\mathrm{A}}(q) \geq \frac{1}{|\mathcal{V}|}\frac{1}{2L}\sum_{v\in\mathcal{V}}\frac{1}{|\mathcal{N}_2(v)|}\sum_{v_2\in\mathcal{N}_2(v)}\|\mathbf{v}-\mathbf{v}_2\|_2^2 + \log\left(\sum_{v_-\in\mathcal{V}}\exp\left(\mathbf{v}^\top\mathbf{v}_-/\tau\right)\right)$$

$$\geq -\frac{1}{|\mathcal{V}|}\frac{1}{2L}\sum_{v\in\mathcal{V}}\frac{1}{|\mathcal{N}_2(v)|}\sum_{v_2\in\mathcal{N}_2(v)}\log\frac{\exp(\mathbf{v}^\top\mathbf{v}_2/\tau)}{\sum_{v_-\in\mathcal{V}}\exp\left(\mathbf{v}^\top\mathbf{v}_-/\tau\right)} \tag{32}$$

$$\geq -\frac{1}{|\mathcal{V}||\mathcal{V}|}\frac{1}{2L}\sum_{v\in\mathcal{V}}\sum_{v_2\in\mathcal{N}_2(v)}\log\frac{\exp(\mathbf{v}^\top\mathbf{v}_2/\tau)}{\sum_{v_-\in\mathcal{V}}\exp\left(\mathbf{v}^\top\mathbf{v}_-/\tau\right)} \triangleq \frac{1}{|\mathcal{V}||\mathcal{V}|}\frac{1}{2L}\mathcal{L}_{\mathrm{SimCLR}}(\mathbf{V}). \tag{33}$$

We can find the last line is exactly the SimCLR-style loss [55] over two-hop graphs. Recent work [59] proves that finding the global optimum of the un-normalized SimCLR-style objective $\mathcal{L}_{\mathrm{SimCLR}}(\mathbf{V})$ is equivalent to solving the matrix factorization problem: $\min_{\mathbf{V}}\|\mathbf{A}_2 - \mathbf{V}\mathbf{V}^T\|_2^2$. Given Equations (31) and (33), we further have:

$$P\left(y_v \neq p_W(\mathbf{v}^*)\right) \leq 2\mathbb{E}_{v,\mathbf{y}_v}\|\mathbf{y}_v - \mathbf{W}\mathbf{v}^*\|_2^2 \tag{34}$$

$$\leq 8\frac{MM}{|\mathcal{V}||\mathcal{V}|}\|(\mathbf{A}_2 - \mathbf{V}^*\mathbf{V}^{*\top})\|_2^2 + 4M^2(1-\hat{h}_2) + 4$$

$$= 8\frac{M^2}{|\mathcal{V}||\mathcal{V}|}\mathcal{L}_{\mathrm{SimCLR}}(\mathbf{V}^*) + 4M^2(1-\hat{h}_2) + 4 + \alpha$$

$$\leq 16M^2L\mathcal{L}_{\mathrm{A}}(q^*) + 4M^2(1-\hat{h}_2) + \beta$$

$$\leq 4M^2(4L\mathcal{L}_{\mathrm{A}}(q) + (1-\hat{h}_2)) + \beta. \tag{35}$$

Here, we convert the mean-squared error bound to classification error bound in Equation (34) as shown by Claim B.9 in [60]. The $\alpha$ is the constant indicating the gap between the optimal loss value of SimCLR-style and matrix factorization losses [59] and $\beta = \alpha + 4$. $\qquad\square$

## C Experimental Details

### C.1 Datasets Details and Statistics

Table 3: Statistics of used homophilic and heterophilic graph datasets in this paper.

| Dataset | #Nodes | # Edges | #Classes | #Features | $\mathcal{H}(\mathcal{G})$ | $\mathcal{H}(\mathcal{G}_2)$ | $\mathcal{S}(\mathcal{G})$ |
|---|---|---|---|---|---|---|---|
| Cora | 2,708 | 5,278 | 7 | 1,433 | 0.81 | 0.71 | 0.89 |
| Citeseer | 3,327 | 4,552 | 6 | 3,703 | 0.74 | 0.56 | 0.81 |
| Pubmed | 19,717 | 44,324 | 3 | 500 | 0.80 | 0.74 | 0.87 |
| Photo | 7,650 | 119,081 | 8 | 745 | 0.83 | 0.66 | 0.91 |
| Computer | 13,752 | 574,418 | 10 | 767 | 0.78 | 0.55 | 0.90 |
| Arxiv | 169,343 | 2,332,386 | 40 | 128 | 0.66 | 0.61 | 0.79 |
| Texas | 183 | 309 | 5 | 1,793 | 0.11 | 0.54 | 0.79 |
| Cornell | 183 | 295 | 5 | 1,703 | 0.30 | 0.40 | 0.40 |
| Wisconsin | 251 | 466 | 5 | 1,703 | 0.21 | 0.42 | 0.42 |
| Chameleon | 2,277 | 36,101 | 5 | 2,325 | 0.23 | 0.35 | 0.67 |
| Squirrel | 5,201 | 216,933 | 5 | 2,089 | 0.22 | 0.22 | 0.73 |
| Crocodile | 11,631 | 360,040 | 5 | 2,089 | 0.25 | 0.30 | 0.71 |
| Actor | 7,600 | 33,544 | 5 | 931 | 0.22 | 0.21 | 0.68 |
| Roman | 22,662 | 32,927 | 18 | 300 | 0.05 | 0.67 | 0.59 |
| Arxiv-year | 169,343 | 1,166,243 | 5 | 128 | 0.22 | 0.58 | 0.77 |

#### C.1.1 One-hop Homophily Levels

We utilize the following edge homophily ratio [15] to measure the one-hop neighbor homophily of the graph. Specifically, the edge homophily ratio $\mathcal{H}(\mathcal{G})$ is the proportion of edges that connect two nodes of the same class:

$$\mathcal{H}(\mathcal{G}) = \frac{\left|\{(u,v) : (u,v)\in\mathcal{E} \wedge y_u = y_v\}\right|}{|\mathcal{E}|} \tag{36}$$

### C.1.2 Neighborhood Context Similarity

To justify our first intuition that two nodes of the same semantic class tend to share similar one-hop neighborhood patterns even in heterophilic graphs, we consider if the same label shares similar one-hop neighborhood distributions of labels in the neighborhoods regardless of the homophily. To measure this property, we calculate the class neighborhood similarity [24] defined as follows:

$$s\left(m, m'\right) = \frac{1}{|\mathcal{V}_m \| \mathcal{V}_{m'}|} \sum_{u \in \mathcal{V}_m, v \in \mathcal{V}_{m'}} \cos(d(u), d(v)), \tag{37}$$

where $M$ is the number of classes, $\mathcal{V}_m$ is the set of nodes with class $m$, and $d(u)$ is the empirical label histogram of node $u$'s neighbors over $M$ classes. $cos(\cdot)$ represents the cosine similarity function. This cross-class neighborhood similarity measures the neighborhood distributions between different classes. When $m = m'$, $s(m, m')$ calculates the intra-class similarity. To measure the neighborhood similarity, we average the intra-class similarities of all classes:

$$\mathcal{S}(\mathcal{G}) = \sum_{m=1}^{M} \frac{1}{M} s(m, m). \tag{38}$$

If nodes with the same label share the same neighborhood distributions, the class neighborhood similarity $\mathcal{H}_s(\mathcal{G})$ is high.

### C.1.3 Two-hop Monophily Levels

To further verify our second intuition in the introduction: two-hop similarities (monophily) still exist even without the one-hop homophily assumption, we also consider monophily properties of two-hop neighborhoods. Following [16], we use the following two-hop homophily to measure the monophily, where the neighborhood of each node is defined to be the nodes of exactly two hops away.

$$\mathcal{H}(\mathcal{G}_2) = \frac{1}{|\mathcal{V}|} \sum_v \frac{1}{|N_2(v)|} \sum_{v_2 \in \mathcal{N}_2(v)} \mathbb{1}_{y_{v_2} = y_v}, \tag{39}$$

where $\mathcal{N}_2(v)$ denotes the set of two-hop neighborhoods of $v$.

The statistics of datasets, including their one-hop and two-hop homophily levels and neighborhood similarities, are given in Table 3. We can observe both homophilic and heterophilic graphs exhibit strong neighborhood similarity calculated based on one-hop neighborhoods. In addition, the two-hop homophily level is generally higher than the one-hop homophily level in heterophilic graphs. This observation confirms our intuition that two-hop similarities still exist without a strong one-hop homophily level. The descriptions of the datasets are given below:

**Cora, Citeseer, and Pubmed** [49]. They are among the most widely used node classification benchmarks. Each dataset is a citation and high-homophily graph, where nodes are documents, and edges are citation relationships from one to another. The class label of each node is based on the research field. A bag of words of its abstracts is used as the features of nodes. The public split is used for these datasets, where each class has fixed 20 nodes for training, another fixed 500 nodes and 1000 nodes for validation/test, respectively for evaluation.

**Computer and Photo** [5, 51]. They are co-purchase graphs from Amazon, where nodes represent products and frequently bought products are connected by edges. Node features represent product reviews, and class labels indicate the product category. Following the experiment settings from [12], we randomly split the nodes into train/validation/test (10%/10%/80%) sets.

For these homophilic datasets, we utilized the process version of them provided by Deep Graph Library [61]. These datasets can be found in `https://docs.dgl.ai/en/0.6.x/api/python/dgl.data.html`.

**Ogbn-arxiv (Arxiv)** [52]. It is a citation network between all Computer Science (CS) arXiv papers. Each node represents one paper and each edge indicates the citation relationships between two papers. The feature of each node is obtained by a 128-dimensional feature vector through averaging the embeddings of words in its title and abstract. The embeddings of words are obtained by running the skip-gram over the MAG corpus. Following [12], we also use 10%/10%/80% split for this dataset.

**Texas, Wisconsin and Cornell** [33]. They are webpage networks collected from computer science departments of different universities by Carnegie Mellon University. For each network, nodes are

web pages and edges indicate hyperlinks between web pages. Node features are bag-of-words representations of web pages. The task is to classify nodes into five categories: student, project, course, staff, and faculty.

**Chameleon, Crocodile and Squirrel** [47]. They are Wikipedia networks, where nodes represent web pages and edges represent hyperlinks between them. Features of nodes are several informative nouns on Wikipedia pages. Labels of nodes are based on the average daily traffic of the web page.

**Actor** [33]. It is an actor-only induced subgraph of the film-director-actor-writer network. Nodes correspond to actors and edges represent the co-occurrence of two nodes on the same Wikipedia page. Node features are keywords in Wikipedia pages. Labels are assigned five categories in terms of words on the actor's Wikipedia.

For **Texas, Wisconsin, Cornell, Chameleon, Crocodile, Squirrel and Actor**, we use the raw data provided by the Geom-GCN [33] with the standard fixed 10-fold split for our experiment. These datasets can be downloaded from: `https://github.com/graphdml-uiuc-jlu/geom-gcn`.

**Roman-empire (Roman)** [48] is a heterophilous graph based on the Roman Empire article in English Wikipedia. Each node in the graph corresponds to one (non-unique) word in the text. The node features are from word embeddings. The class of a node is its syntactic role (the 17 most frequent roles as unique classes and all the other roles are grouped into the 18th class). Following [48], we utilize the fix 10 random 50%/25%/25% train/validation/test splits. This dataset can be found in `https://github.com/yandex-research/heterophilous-graphs`.

**Arxiv-year** [16] is a citation network from a subset of the Microsoft Academic Graph, where the task focuses on predicting the year that a paper is posted. Nodes are papers, and edges are relevant to citations. The node features correspond to the average of word embeddings of the title and abstract of the papers. Following [16], 50%/25%/25% train/val/test split is utilized for this dataset. This dataset can be found in `https://github.com/CUAI/Non-Homophily-Large-Scale`.

## C.2 Baselines

**LINE** [14]: This network embedding model proposed an objective function to preserve both the first-order and second-order proximities.

**VGAE** [9]: VGAE is a generative model based on variational autoencoder for node representation learning by directly reconstructing adjacency matrix.

**DGI** [2]: It is a self-supervised learning method that maximizes the mutual information between node representations and graph summary.

**GCA** [4]: This is a graph contrastive representation learning method with adaptive augmentation that incorporates various priors for topological and semantic aspects of the graph.

**BGRL** [5]: This model is a graph contrastive learning method, where the alternative augmentations of the graph are predicted to learn representations of nodes.

**CCA-SSG** [6]: CCA-SSG is a graph contrastive learning model, which encourages the learned representations of nodes by reducing the correlation between different views.

**L-GCL** [12]: It is a self-supervised node representation learning method, which samples positive samples from the first order neighborhoods and kernelizes negative loss to reduce the training time.

**SP-GCL** [53]: This is a single-pass graph contrastive learning method based on the concentration property of node representations.

**HGRL** [29]: This is a self-supervised representation learning framework on graphs with heterophily, which leverages the node original features and the high-order information.

**DSSL** [18]: This method introduces a representation learning framework by decoupling the diverse neighborhood context of a node in an unsupervised manner.

## C.3 Setup and Hyper-parameter Settings

We use official implementation publicly released by the authors on Github of the baselines. For fair comparison, we used grid search to find the best hyperparameters of the baselines. We run

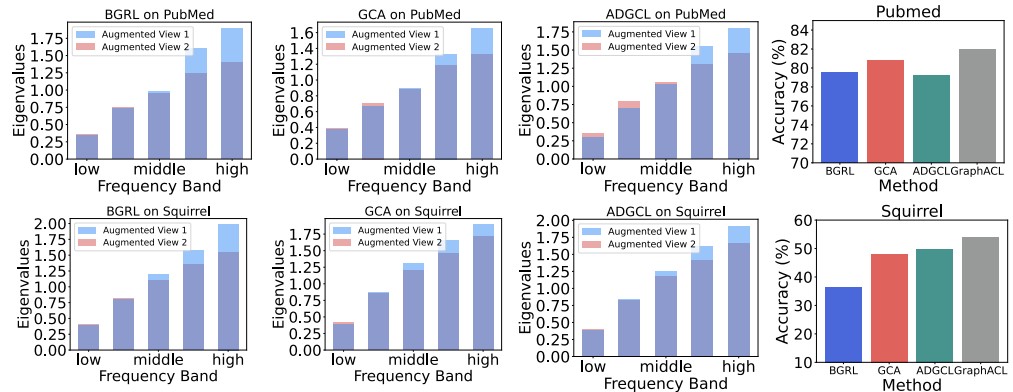

Figure 7: The average spectrum changes across these frequency bands for two augmented views. The perturbations resulting from augmentations reveal a consistent trend in the graph spectrum.

experiments on a machine with a NVIDIA RTX A6000 GPU with 49GB of GPU memory. In all experiments, we use the Adam optimizer [62]. A small grid search is used to select the best hyperparameter for all methods. In particular, for our GraphACL, we search $\lambda$ from {0, 0.90, 0.95, 0.97, 0.99, 0.999, 1}, $D$ from {256, 512, 1024, 2048, 4096, 8192}, $\tau$ from {0.25, 0.5, 0.75, 0.99, 1}, and $K$ from {1, 5, 10} when we utilize the negative sampling. We tune the learning rate over {0.001, 0.0005, 0.0001} and weight decay over {0, 0.0001, 0.0003, 0.000001}. We select the best configuration of hyper-parameters based on average accuracy only based on the validation set.

Table 4: Node clustering performance in terms of NMI (%) on homophilic and heterophilic graphs

| Method | Citeseer | Computer | Photo | Arxiv | Texas | Cornell | Squirrel | Arxiv-year |
|---|---|---|---|---|---|---|---|---|
| VGAE | 36.40±0.01 | 22.30±0.00 | 53.00±0.04 | 31.10±0.01 | 27.75±0.16 | 17.87±0.13 | 10.83±0.09 | 25.64±0.05 |
| DGI | 43.90±0.00 | 31.80±0.02 | 47.60±0.03 | 31.90±0.01 | 34.17±0.07 | 15.92±0.15 | 8.49±0.13 | 24.35±0.08 |
| BGRL | 45.38±0.04 | 36.20±0.04 | 54.61±0.08 | 33.71±0.09 | 33.59±0.15 | 19.74±0.14 | 15.13±0.09 | 25.31±0.05 |
| DSSL | 45.91±0.06 | 36.82±0.08 | 54.99±0.05 | 32.98±0.06 | 38.22±0.15 | 20.36±0.08 | 19.85±0.13 | 24.45±0.07 |
| L-GCL | 46.52±0.08 | 37.51±0.09 | 55.25±0.01 | 32.77±0.05 | 37.92±0.10 | 19.25±0.05 | 18.15±0.11 | 26.87±0.09 |
| GraphACL | 46.82±0.07 | 39.51±0.09 | 55.97±0.04 | 34.15±0.06 | 43.31±0.08 | 22.79±0.05 | 23.96±0.09 | 29.84±0.08 |

## C.4  The Spectrum Changes in Augmented Views

In Figure 7, we categorize the eigenvalues (spectrum) of the symmetric normalized graph Laplacian into five distinct frequency bands, ranging from low to high. Subsequently, we computed the average spectrum changes across these frequency bands for two augmented views. The perturbations resulting from augmentations reveal a consistent trend in the graph spectrum. After augmenting the graph, we can observe that the perturbations in the low-frequency components (far left) are comparatively smaller than those observed in the high-frequency components (far right) for both homophilic (PubMed) and heterophilic (Squirrel) graphs. This observation confirms that mainstream GCL methods can not work well on heterophilic graphs, particularly when the perturbation in high-frequency signals is more significant.

## C.5  Node Clustering Performance

We also conduct node clustering to evaluate the quality of the learned node representations. Specifically, we obtain node representation with GraphACL, then perform k-means clustering on the obtained representations and set the number of clusters to the number of ground truth classes. The experiment is conducted five times. We report the average normalized mutual information (NMI) for clustering in Table 4 on both homophilic and heterophilic graphs. From the table, we can observe that our GraphACL can consistently improve node clustering performance compared to the state-of-the-art self-supervised learning baselines on eight datasets. This observation, along with the results of the node classification, demonstrates the effectiveness of GraphACL in learning more expressive and robust node representations for various downstream tasks. These results further validate that

Table 5: Experimental results (%) on the graph classification results.

| Method | LINE | VGAE | GraphCL | L-GCL | DSSL | GraphACL |
|---|---|---|---|---|---|---|
| **MUTAG** | $75.6_{\pm2.3}$ | $84.4_{\pm0.6}$ | $86.8_{\pm1.3}$ | $85.3_{\pm0.5}$ | $\underline{87.2_{\pm1.5}}$ | $\mathbf{89.4_{\pm2.0}}$ |
| **PROTEINS** | $61.1_{\pm1.7}$ | $74.0_{\pm0.5}$ | $\underline{74.4_{\pm0.5}}$ | $72.9_{\pm0.6}$ | $73.5_{\pm0.7}$ | $\mathbf{75.3_{\pm0.5}}$ |

modeling one-hop neighborhood pattern and two-hop monophily similarity benefit downstream tasks on real-world graphs with various homophily ratios.

## C.6 Graph Classification Performance

Existing works for heterophilic graphs typically focus on node-level tasks (node classification and node clustering). Thus, an empirical comparison of the graph-level task is relatively not straightforward. Nevertheless, for graph classification, we can use a non-parameterized graph pooling (readout) function, e.g., MeanPooling, to obtain the graph-level representation. For graph classification, we conduct experiments on three graph classification benchmarks: MUTAG and IMDB-B, and follow the same setting in GraphCL [3]. The results are shown in Table 5. From the table, we can observe that our ACL framework can also work well on the graph classification task and still can achieve better (or competitive) performance compared to baselines.

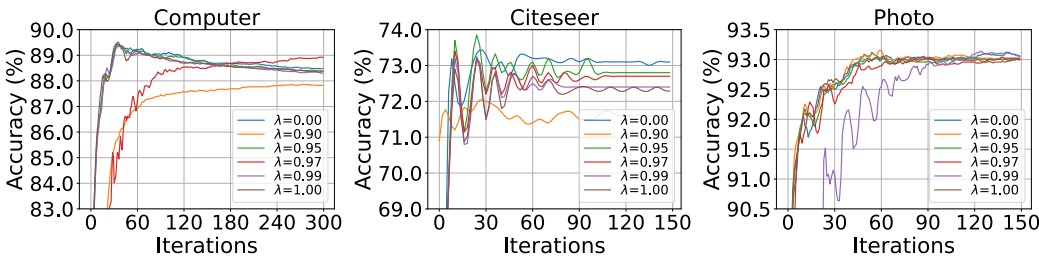

Figure 8: The classification performance curves with varying decay rate $\lambda$ on homophilic graphs.

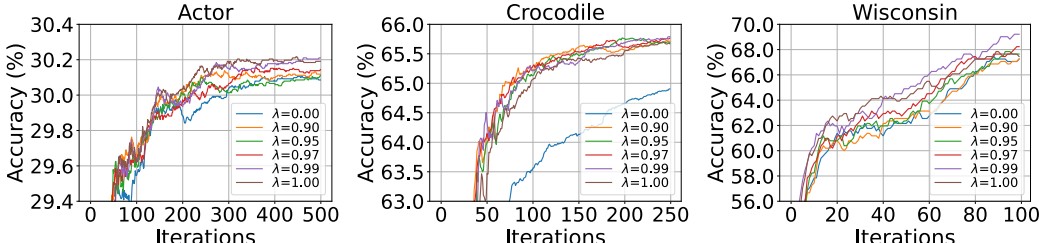

Figure 9: The node classification performance with varying decay rate $\lambda$ on other heterophilic graphs.

## C.7 The Effect of Target Decay Rate

Figures 8 and 9 show the results of node classification with varying decay rate $\lambda$ on four other homophilic and four heterophilic graphs, respectively. We can still observe that having large values of $\lambda$ improves the overall performance. We observe that GraphACL obtains a competitive but not the best result when $\lambda = 0$, which confirms that slowly updating the target network is crucial in obtaining superior performance. We also notice that we can achieve good performance even with a random target encoder, i.e., $\lambda = 1$ on some datasets, which can be explained by the recent work [63] that shows only using the simple stop-gradient operation can sometimes prevent collapsing.

## C.8 The Effect of Temperature

Figures 10 and 11 show the results of node classification with variable temperature $\tau$ in homophilic and heterophilic graphs. We can observe that our GraphACL is not very sensitive to temperature $\tau$ on heterophilic graphs, while moderate hardness of the softmax (large $\tau$) produces the best result. For homophilic graphs, the large or small temperature can lead to poor performance.

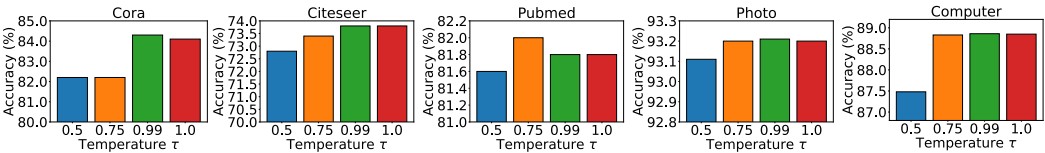

Figure 10: The effect of temperature $\tau$ on five homophilic graphs.

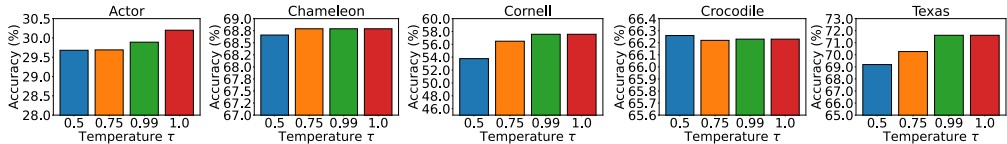

Figure 11: The effect of temperature $\tau$ on five heterophilic graphs.

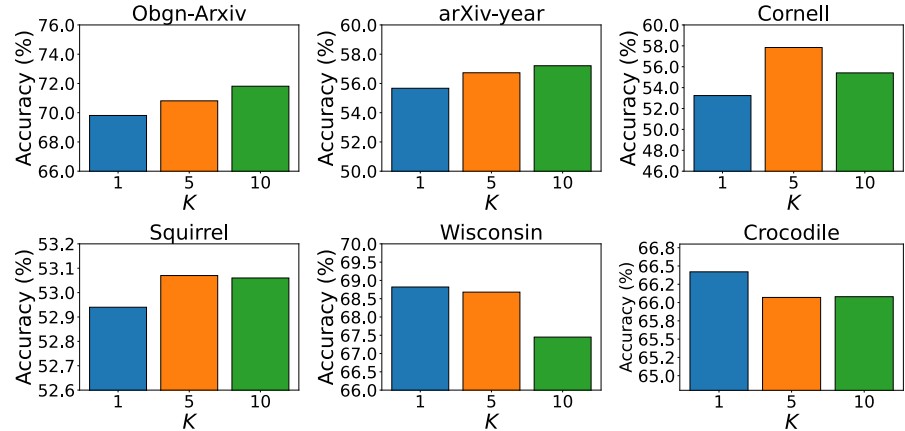

Figure 12: The effect of the number of negative samples $K$.

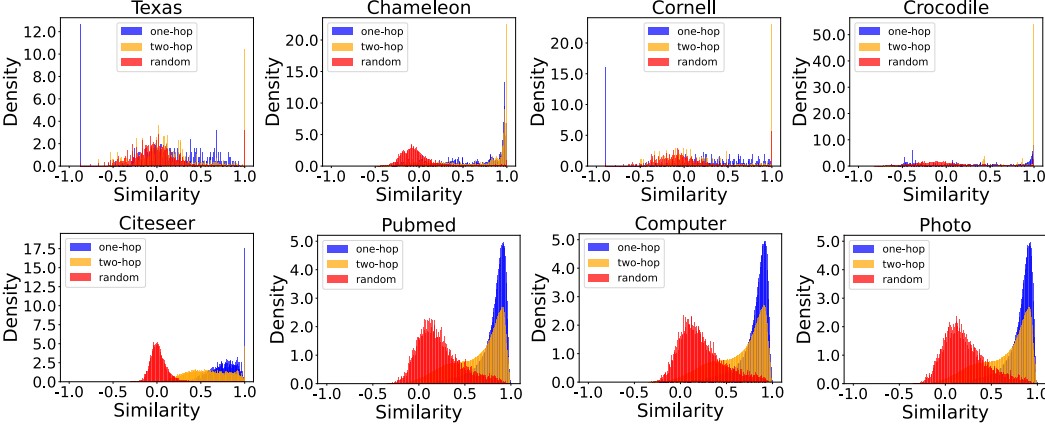

Figure 13: The distribution of pair-wise cosine similarity calculated by learned representations on randomly sampled node pairs, one-hop neighbors and two-hop neighbors.

## C.9 The Effect of the Number of Negative Samples

We run a sweep over the size of negative samples $K$ to study how it affects performance. We vary $K$ as $\{1, 5, 10\}$. For each $K$, we first learn node representation and then use the learned node representation for node classification. Figure 12 shows the results with varying $K$. From the figure, we can observe that a small number of negative samples (e.g., $K = 5$) is enough to achieve good performance on all graphs. For homophilic graphs, we can observe that large $K$ can promote the

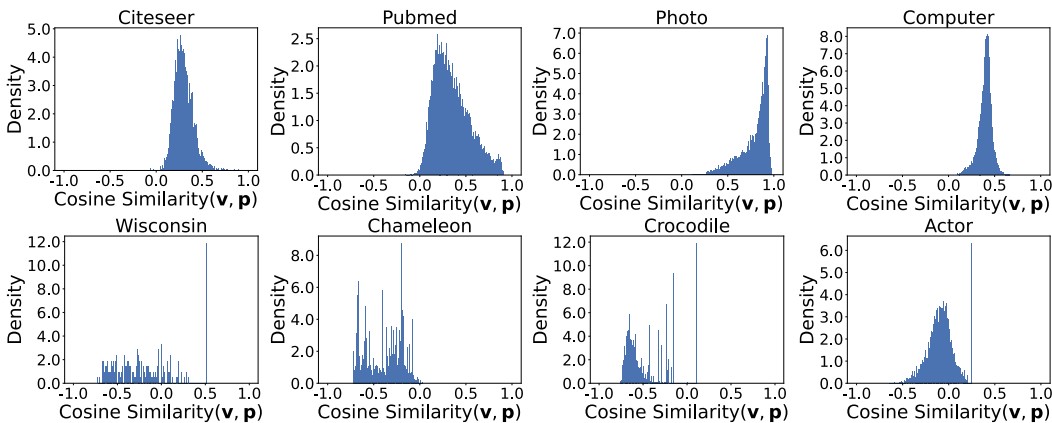

Figure 14: Similarity (cosine similarity between representation $\mathbf{v}$ and prediction $\mathbf{p} = g_\phi(\mathbf{v})$) histogram on different graphs.

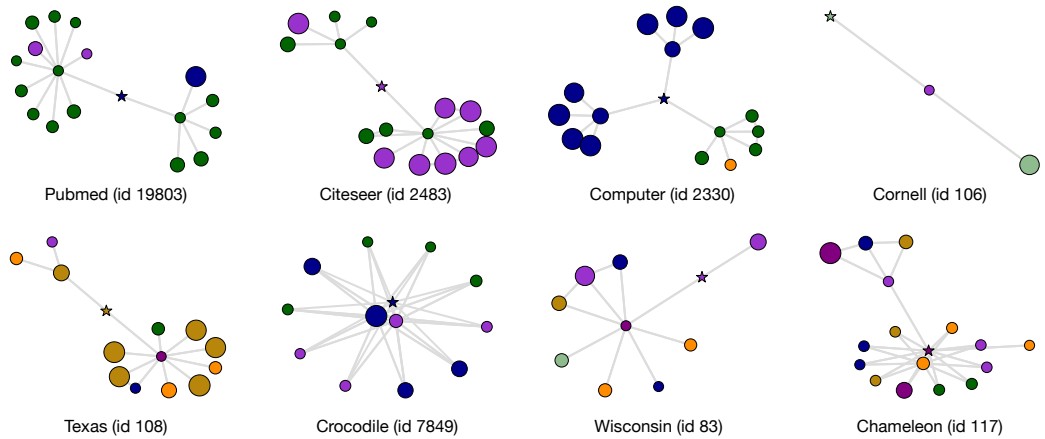

Figure 15: Case studies. Node colors denote the semantic labels of nodes. The size of the node is proportional to its similarity to the central node denoted as the star.

performance of GraphACL. In contrast, for heterophilic graphs, training with large $K$ will lead to a slight drop in performance. A possible reason is that the randomly sampled negative samples can not represent the whole node-set, given the heterogeneous and diverse patterns of heterophilic graphs.

### C.10  More Similarity Histograms of Node Pairs

Figure 13 shows the additional results on the representation similarity. The observations are generally similar to the results in the main body of the paper. As shown by the figure, we can observe that the randomly sampled node pairs are easier to be distinguished from one-hop and two-hop neighbors based on the representation similarity, which demonstrates that our GraphACL indeed captures the semantic meaning of node and will desirably push away the semantically dissimilar nodes. Moreover, the two-hop similarities in heterophilic graphs are much larger than that in homophilic graphs. These phenomena provide explanations for why GraphACL achieves good performance by capturing two-hop monophily structure information. Figure 14 shows the cosine similarity between the $\mathbf{v}$ and prediction $g_\phi(\mathbf{v})$ for each node on other graphs. We can find that the cosine similarities are typically smaller than 1, which shows that the predictor $g_\phi$ is not an identity matrix after convergence. Moreover, we can observe that, compared to the homophilic graph Cora, the similarities on the heterophilic graph Squirrel are smaller. Since the objective of GraphACL will pull $\mathbf{p} = g_\phi(\mathbf{v})$ and $\mathbf{u}$ together, thus GraphACL can automatically differ identity representation $\mathbf{v}$ and preference representation $\mathbf{u}$, which is important for modeling heterophilic graphs. For the homophilic graph

Cora, the node identity representation **v** and the preference representation **u** should be similar, which is also captured by our simple contrastive objective in GraphACL.

### C.11 Additional Case Study Results

In Figure 15, we provide the additional case studies and visualization results on other graphs. We can find that, in most cases, nodes sharing the same semantic classes with the central nodes have larger similarities to the central nodes. This observation interprets the reason why GraphACL can achieve good performance. We observe that GraphACL can effectively capture the local neighborhood pattern and two-hop monophily for both homophilic and heterophilic graphs, which empirically verifies our theoretical analysis given in the main body of the paper.

## D  Societal Impacts and Limitations

There are numerous graphs in the real world that exhibit heterophilic properties, such as transaction networks, ecological food networks, and molecular networks, in which the connected nodes possess dissimilar features and distinct class labels. Given the successful deployment of Graph Neural Networks (GNNs) in various human-related real-world applications, including social networks, knowledge graphs, and molecular property prediction, it is crucial to propose unsupervised representation learning techniques for GNNs that can effectively handle both homophilic and heterophilic graphs, thereby potentially yielding direct social impacts.

The collection of labeled data is often expensive and impractical, particularly in domains requiring specialized knowledge, such as medicine and chemistry. Considering the potential positive impact, we believe that our work can assist researchers and practitioners in devising solutions that alleviate the reliance on labeled data. However, it is important to acknowledge the potential negative consequences, as these learned representations could also be exploited for malicious purposes, such as adversarial attacks on GNNs that exploit systematic biases. Nonetheless, we believe that our simple framework and theoretical insights derived from this work contribute as a small step towards advancing the simplicity and generalizability of graph contrastive learning models within the research community.

