# OpenReview forum: "Simple and Asymmetric Graph Contrastive Learning without Augmentations"
_NeurIPS.cc/2023/Conference — NeurIPS 2023 poster_

### Official Review · Reviewer_gZUY · 2023-06-15

**Soundness:** 3 good
**Presentation:** 3 good
**Contribution:** 3 good
**Rating:** 6
**Confidence:** 4

**Summary:**

This paper proposes an asymmetric contrastive learning framework for the homophilic and heterophilic graphs, which does not rely on graph augmentations and homophily assumptions. The theoretical analysis and empirical results further support the effectiveness of the proposed method.

**Strengths:**

1. This paper is easy to understand and the motivation is clear and reasonable.
2. The proposed method is simple yet effective. The authors further demonstrate the effectiveness of the proposed method from the information theory and downstream tasks perspectives. Such results give a theoretical understanding of this work.
3. Extensive experimental results including comparison experiments, ablation study, visualization, and case study empirically verify the effectiveness of the proposed method. Moreover, the proposed method shows superior performance on both the  homophilic and heterophilic graph datasets.


**Weaknesses:**

1. This paper is somewhat over-claim and some important references are missed. In lines 45-47, the authors claim that this work makes the first attempt to design contrastive learning objectives by neither relying on explicit nor implicit homophily assumptions. Actually, there are several works already done. For example,
[1] Wang H, Zhang J, Zhu Q, et al. Augmentation-free graph contrastive learning. arXiv preprint arXiv:2204.04874, 2022.

2. The proposed objective function seems like a variant of the GCL with representation smoothing diagram and the GCL with augmented views diagram. Actually, the difference between them and the proposed method is the definition of the positive pairs.

3. In this work, the authors claim that the proposed method does not rely on the homophily assumption and conduct a theoretical analysis to further support the proposed method. Are these theorems hold without any assumptions?

**Questions:**

See above

---

> ### Author Rebuttal · Authors · 2023-08-10
>
> Dear reviewer gZUY, we thank you for your valuable suggestions and positive feedback. **We are happy to hear that you found our paper to be well-written and strong in both empirical and theoretical aspects.** The following is our point-to-point response to your comments:
>
> **(C1). This paper is somewhat over-claim and the important references [1] are missed.**
>
> **(R1).** We appreciate your insightful comments! **We believe there are some important misunderstandings:**
>
> - We would like to clarify that **we did not miss the reference [1]. We cited this reference as [48] and also empirically compared their method in our experiments. It is essential to note that [1] and [48] refer to the same model and same experimental results from the same authors, with the titles being different.**
>
> - In addition, the model in [1] (i.e., [48]) actually cannot outperform many recent GCL methods on many homophilic datasets (Cora, CiteSeer, PubMed, and Photo), as reported in their paper. Conversely, our GraphACL achieves state-of-the-art results on those homophilic datasets. Thus, our claim that we are the first to attempt answering the question, "What kind of insights and contrastive learning objectives should we look for, in order to ensure good node representations on **both homophilic and heterophilic graphs**", **is indeed valid and not an over-claim**. We appreciate your comments and will ensure that these points are clearly stated in the revision.
>
> **(C2). The proposed objective function seems like a variant of the GCL with representation smoothing diagram and GCL with augmented views diagram. Actually, the difference between them and the proposed method is the definition of the positive pairs.**
>
> **(R2).** Thank you for your comments. While GraphACL involves positive pairs, our contributions and differences compared to other contrastive schemas go beyond that. Here, we will provide a detailed clarification:
>
> - *Comparisons with GCL methods with representation smoothing:* Different from this scheme which is based on the homophily assumption, GraphACL is asymmetric contrastive learning induced by a simple asymmetric predictor. **We theoretically and empirically show that this simple asymmetric predictor can jointly capture one-hop heterophilic patterns and two-hop monophily, which are important for heterophilic graphs.**
>
> - *Comparisons with GCL methods with augmented views:* This contrastive learning scheme relies on graph augmentations, and are built based on the idea that the augmentation can preserve the semantic nature of samples, i.e., the augmented samples have invariant semantic labels with the original one. However, our GraphACL is not based on augmentations, but  a simple asymmetric predictor. GraphACL directly considers analyzing neighborhood distribution in the original graph. **Thus, our GraphACL design is inherently different from GCL methods with augmented views.** Moreover, We theoretically show that GraphACL also implicitly aligns the two-hop neighbors and enjoys a good downstream performance for both homophilic and heterophilic graphs, which is theoretically and intuitively unclear for GCL with augmented views.
>
> Given the above, our studied problems and insights are fundamentally different from these two schemes. Our theoretical analyses are novel and have not been proposed by these two GCL schemes. We believe that our work offers unique insights, and our technical contributions remain significant, when compared to the current GCL schemes.
>
> **(C3). In this work, the authors claim that the proposed method does not rely on the homophily assumption and conduct a theoretical analysis to further support the method. Are these theorems hold without any assumptions?**
>
> **(R3).** Thank you for your insightful questions! Actually, our theorems 1, 2, and 4 do not rely on any assumptions. Only theorem 3 makes some lightweight, reasonable, and widely-used assumptions, as mentioned in our paper. Not surprisingly, some assumptions are necessary for theoretical analysis in generalization for unsupervised node representation learning [2, 3, 4]. Specifically, our theorem 3 (downstream error on learned representation) assumes that the used downstream classifier is a mean classifier. This mean classifier assumption is very mild and has been utilized by many works [2, 3, 4]. Another lightweight assumption mentioned in our paper is the balanced class distributions. This assumption is also mild and widely-used in the literature [5]. Moreover, our theorem 3 can be easily extended to unbalanced settings in the future by considering a label shift term, as shown in domain adaptation literature [6]. Moreover, GraphACL consistently outperforms the baselines even when the classes are not well balanced, indicating that GraphACL is robust to violations of the class balance assumption.
>
> **In light of these responses, we sincerely hope our posted responses have addressed your comments and clarified any misunderstandings. We believe your comments can be easily addressed in the final version and genuinely hope that you could consider increasing your score. If you have any notable points of concern that remain unaddressed, please do share them with us, and we will promptly address them. Thank you for your efforts!**
>
> [1] Augmentation-free graph contrastive learning. arXiv preprint arXiv:2204.04874, 2022
>
> **[48] (the cited number in our submission)** Can Single-Pass Contrastive Learning Work for Both Homophilic and Heterophilic Graph? arXiv preprint arXiv:2211.10890, 2022
>
> [2] A Theoretical Analysis of Contrastive Unsupervised Representation Learning. ICML 2019
>
> [3] Understanding Negative Samples in Instance Discriminative Self-supervised Representation Learning. NeurIPS 2021
>
> [4] On the Surrogate Gap between Contrastive and Supervised Losses. ICML 2022
>
> [5] Debiased Contrastive Learning. NeurIPS 2020
>
> [6] Domain Adaptation with Conditional Distribution Matching and Generalized Label Shift. NeurIPS 2020

---

> > ### Comment · Reviewer_gZUY · 2023-08-15
> > **Response to the rebuttal**
> >
> > Thanks for your efforts, I will maintain my positive score.

---

> > > ### Author Response · Authors · 2023-08-15
> > > **Thank you for your comments!**
> > >
> > > Thank you very much for your positive comments! We sincerely appreciate your helpful feedback and are grateful for your approval.
> > >
> > > Best wishes,
> > >
> > > Authors

---

### Official Review · Reviewer_cxpZ · 2023-07-02

**Soundness:** 3 good
**Presentation:** 3 good
**Contribution:** 3 good
**Rating:** 6
**Confidence:** 4

**Summary:**

This work first points out that existing GCL can fail to generalize to heterophilic graphs, then develops a new framework called GraphACL based on an encoder capturing one-hop neighbourhood context and two-hop monophyly. Experiments validate the effectiveness of the proposed method.

**Strengths:**

The paper is well organized. Each part has a clear goal and supports the authors' claim well.

The proposed method, Graph Asymmetric Contrastive Learning, is not only explained in detail but also justified by solid theoretical analysis. The authors adequately answered how and why it works.

In the empirical study, the choice of baselines has good coverage. The selection of datasets respects the theme of each subsection, and the experiment setup is reasonable. The experiments are well explained, making it easy to reproduce them.

**Weaknesses:**

There are no significant weaknesses in this paper. The application scenario of the proposed method may be a little bit limited, but this is completely acceptable.

**Questions:**

Does GADC have the potential to be applied to graph tasks other than node classification shown in the paper?

**Limitations:**

The authors adequately addressed their work's limitations and potential societal impact.

---

> ### Author Rebuttal · Authors · 2023-08-10
>
>  Dear reviewer cxpZ, **thank you for the great summarization of our contributions on both theoretical  and empirical analysis, and we appreciate your very positive and encouraging comments.** Please see our responses below:
>
> **(C1). There are no significant weaknesses in this paper. The application scenario of the proposed method may be a little bit limited, but this is completely acceptable.**
>
> **(R1).** Thanks for your positive comments! **We would like to further clarify that the application scenario of our method is actually wide due to the following two important reasons:**
>
> -  (1) Heterophilic graphs are important in various real-world domains, making them worth studying and essential to understand. Many real-world graphs demonstrate heterophilic properties. For example, in online transaction networks, fraudsters tend to connect with customers rather than other fraudsters [1]. In molecular networks, protein structures often consist of different types of amino acids linked together [2, 3]. Recent work [3] also provides a comprehensive review of so many GNNs for heterophilic graphs. High-quality datasets covering various heterophilic real-world applications have been made available through recent work [4]. For instance, malicious node detection, an important application of graph machine learning, is known to be heterophilic in many settings [4]. **Thus, studying heterophilic graphs is a significant research problem with benefits for many applications involving them.**
> -  (2) Learning effective representations for both homophilic and heterophilic graphs is also crucial for various real-world applications. In many scenarios, **collecting labeled data can be expensive and impractical, especially when domain knowledge is required, as in medicine and chemistry [5, 6].** Our model, as a key simple yet effective representation learning method, will show great potential in various high-level applications, including molecular property prediction [5,6], molecular graph generation, and drug-drug interaction prediction [7].
>
>
> **(C2). Does GraphACL have the potential to be applied to graph tasks other than node classification shown in the paper?**
>
> **(R2).** **Yes, actually, we have already conducted experiments on graph classification tasks. The reviewer can find the results in Table 8 in our Appendix D.5.** From the table, we can observe that our GraphACL performs well on the graph classification task and achieves better (or competitive) performance compared to baselines, which further strengthens our contribution.
>
> **We appreciate the efforts from the reviewer and also sincerely hope our posted responses can address your questions. We also believe your comments can also be easily addressed in the revision. As noticed by the reviewer, our work is  simple yet effective, and provides extensive theoretical and experimental analysis. In light of these responses, we sincerely hope you could consider increasing your score.  Please feel free to let us know if there are any remaining questions. Thank you for your efforts!**
>
>
> [1] NetProbe: A Fast and Scalable System for Fraud Detection in Online Auction Networks. WWW 2007
>
> [2] Beyond Homophily in Graph Neural Networks: Current Limitations and Effective Designs. NeurIPS 2020
>
> [3] Graph Neural Networks for Graphs with Heterophily: A Survey. ArXiv preprint
>
> [4] Large Scale Learning on Non-Homophilous Graphs: New Benchmarks and Strong Simple Methods. NeurIPS 2021
>
> [5] Self-supervised Graph Transformer on Large-scale Molecular Data. NeurIPS 2020
>
> [6] Motif-based Graph Self-supervised Learning for Molecular Property Prediction. NeurIPS 2021
>
> [7] A Systematic Survey of Chemical Pre-trained Models. In Arxiv.

---

> > ### Comment · Reviewer_cxpZ · 2023-08-15
> >
> > Thanks for the clarification, and I appreciate the authors' effort in this. I will maintain my positive score.

---

> > > ### Author Response · Authors · 2023-08-15
> > > **Thank you for your update!**
> > >
> > > Thank you very much for your positive comments! We sincerely appreciate your helpful feedback and are grateful for your approval.
> > >
> > > Best wishes,
> > >
> > > Authors

---

### Official Review · Reviewer_ycWA · 2023-07-06

**Soundness:** 3 good
**Presentation:** 3 good
**Contribution:** 2 fair
**Rating:** 6
**Confidence:** 3

**Summary:**

This paper presents GraphACL, which aims to tackle limitations of other graph contrastive learning works which have implicit or explicit homophily assumptions, and suffer in learning effective representations for heterophilic tasks.  The approach is designed to leverage the principle of monophily, and the authors evaluate their work on several homophilic and heterophilic datasets, achieving at-par performance on the former and improved performance on the latter, on downstream node classification tasks.

**Strengths:**

- The approach presented here is intuitive once the monophilic principle is understood, and the implementation seems straightforward.

- Results suggest the approach is strongly effective (e.g. Table 1), and hte improvement on heterophilic datasets without any material performance loss / delta on homophilic datasets is compelling.

- The paper is well-written, and the figures help clarify the principles motivating the design.

**Weaknesses:**

- Lines 69-80 could greatly benefit from a toy example with nodes illustrated; it is confusing to understand the principles of monophily and the design of GraphACL without such an example early in the paper.

- "we are faced with the challenge of simultaneously capturing the one-hop neighborhood context and monophily in the contrastive objective" (line 74) -- it's not clear why this is a particular challenge; is there something about capturing these two together that is difficult to reconcile in an objective?  If so, I didn't understand this from the text.

- Lines 40 and 97 rely quite a bit on the (in paper) reference [22] to reference issues with contrasting view-based GCL approaches on heterophilic graphs; it would be helpful to include some more formal claims or concrete examples in this work to make it more self-contained, since the work strongly seems to draw motivation from [22]'s findings.  Without this, it is difficult to fully appreciate the supposed homophilic biases in those GCL approaches.

- The section in Related Work from line 137 onwards seems to categorize BGRL as a contrastive method.  BGRL is not conventionally contrastive due to its lack of utilizing negative examples.  There are other such non-contrastive methods (several mentioned in [1]) which are used in graph SSL which should probably be mentioned in the related work in this paper.  In fact, I believe the authors' intent to pursue a graph SSL approach which does not use augmentations but instead uses induced asymmetry is actually the objective of many "non-contrastive" methods.  I would encourage they look into the literature and evaluate whether they indeed think their method is contrastive or not, given it's lack of need of negative samples.

- The words "context" and "preference" are used quite often throughout the paper (I think sometimes to mean the same thing).  It would be helpful to formalize these words and just use the same word frequently, as it can be confusing to disambiguate which representation is which.

- Can the authors clarify if/why the loss in Eqn 4 is required?  Methods like SimSiam, BYOL, BGRL etc. which use this kind of predictor structure and stop-gradient show they avoid collapse through the asymmetric predictor and the EMA procedure on target weight update.  Does GraphACL not naturally avoid collapse for the same reasons?  Does it require this uniformity regularization?

- Figure 3 could go earlier to help introduce monophily and the intuition behind what existing GCL methods prioritize and what this method prioritizes.

- There are a few very strong self-supervised GNN approaches missing from baselines, e.g. [2] or the approach mentioned in [1].

- There are opportunities to better evaluate the quality of self-supervised representations using link-level tasks (e.g. link prediction), or several of the tasks mentioned in [2], which may offer improved understanding beyond node classification and node clustering (referenced in Sec 6.1)

- Evaluation on some larger datasets with greater data diversity and lower homogeneity would have been more compelling, e.g. ogbn-arxiv or ogbn-products.  Flickr might also be a useful moderate-size heterophilic dataset to try out.

- The node clustering results should be included in the main paper in my opinion -- the evaluation on just 1 downstream SSL task is fairly "light" and could use greater support to evidence that GraphACL is a generally strong SSL method, rather than "SSL for node classification" method.  This is one of the biggest gripes in this paper -- the results demonstrating that GraphACL is a very effective SSL method are sparse, with only 1 demonstrated task in the main (evaluated) paper in terms of node classification. Other graph SSL work typically considers several downstream tasks and evaluates across them.

[1] Link Prediction with Non-Contrastive Learning (Shiao et al, ICLR'23)
[2] Multi-task Self-supervised Graph Neural Networks Enable Stronger Task Generalization (Ju et al, ICRL'23)

**Questions:**

Please see "Weaknesses" for general comments / concerns which would be helpful during rebuttal.

**Limitations:**

These are addressed in Appendix E.

---

> ### Author Rebuttal · Authors · 2023-08-10
>
> Dear reviewer ycWA, **we appreciate your perception that our model is implementation-wise simple, strongly effective, and intuitive. We thank your insightful comments and give our responses below:**
>
> **(C1). Lines 69-80 could greatly benefit from a toy example**
>
> **(R1).** Thanks for your great suggestion! As mentioned in your following comments, we will move toy examples in Figure 3 to Lines 69-80.
>
> **(C2). It's not clear why this is a particular challenge (line 74)**
>
> **(R2).** Thanks for your insightful question! We apologize for your confusion and will clearly state the following in the revision. Reconciling the objective of capturing both the one-hop neighborhood (heterophilic) context and monophily (two-hop similarity) in one contrastive objective is challenging. For instance, although homophily-driven objectives can indirectly induce monophily by directly encouraging one-hop similarity, they neglect the heterophilic context, where one-hop connected nodes are not similar. Thus, our GraphACL focuses on addressing the challenges of simultaneously and theoretically capturing one-hop neighborhood (heterophilic) context and monophily in one simple contrastive objective.
>
> **(C3). …issues with contrasting view-based GCL approaches on heterophilic graphs... include some concrete examples to make it more self-contained…**
>
> **(R3).**  Thanks for your suggestion! Please see our detailed responses in **Global Response 2**.
>
> **(C4). ...whether they indeed think BGRL and [1]  are contrastive or not, given the lack of need for negative samples...**
>
> **(R4).** Thanks for your insightful comments! The BGRL and [1] should definitely be viewed as non-contrastive due to the lack of negative samples. Our intention in Lines 142-143 is to say that BGRL is a graph self-supervised learning method based on augmentations. We will clearly state this in the main text. Moreover, as mentioned in our paper (section 4.1), even though they also use an asymmetric projection head, our motivation, objective, and theoretical insights differ significantly from BGRL which is based on invariant augmentation assumption.
>
> **(C5). It would be helpful to formalize  "context" and "preference" and just use the same word frequently**
>
> **(R5)**. Thanks for your suggestion! Please see our detailed responses in **Global Response 1**. We agree and will utilize the same "preference" word.
>
> **(C6). Does GraphACL not naturally avoid collapse compared to SimSiam, BYOL, and BGRL  for the same reasons? Does it require this uniformity regularization?**
>
> **(R6).** Thanks for your very insightful questions! Uniformity is important and required for GraphACL, especially for homophily graphs. As shown in our ablation studies, GraphACL w/o uniformity loss will lead to a performance drop instead of completely collapsed solutions since it still serves as a strong baseline. Moreover, the ablation (w/o both asymmetric predictor and uniformity loss) also serves as a valid baseline and does not lead to a completely collapsed solution. This is contrary to SimSiam, BYOL, and BGRL, which focus on predicting the sample representation in one augmented view via the same sample from another view. One possible reason is that GraphACL is not based on augmentation but tries to predict the representations of neighbors via an asymmetric predictor. Thus, the observed graph structure serves as a strong prior and inductive bias. Since neighbors in the graph, even in homophily graphs, do not always share the same class or similar features, making GraphACL hard to completely collapse.
>
> We leave the theoretical and deeper understanding of this phenomenon for future work, as it may extend beyond the scope of a single paper. Nonetheless, as demonstrated in our experiments, adding a uniformity loss remains crucial for enhancing representation diversity and increasing inter-class variation, ultimately leading to good generalization. We will explicitly state this in the main text.
>
> **(C7). Fig. 3 could go earlier to help introduce monophily**
>
> **(R7).** Thank you! We agree and will move Fig. 3 earlier into introduction.
>
> **(C8). A few very strong self-supervised GNN approaches [1,2] missing from baselines**
>
> **(R8).** Thanks for raising the great works [1,2]! Following your suggestions, we compare GraphACL with [1,2]. Please see details and results in **Global Response 3.**
>
> **(C9). Beyond node classification and node clustering and using link-level tasks or the tasks in [2]**
>
> **(R9)**. Thanks for your suggestions! **We want to kindly remind the reviewer that we already have not only evaluated node classification (Table 1) and node clustering (Table 5), but also the graph classification (Table 8).** Nevertheless, we agree that including more tasks will be better. Thus, we include two more tasks: link prediction and partition prediction. For simplicity,  we follow the same settings as [2] as it provides splits for heterophilous graphs. The results given in attached PDF (Table 2) in Global Response show that GraphACL can still achieve better (or competitive) performance compared to elaborate methods.
>
> **(C10). Evaluation on some larger datasets, e.g. ogbn-arxiv or ogbn-products**
>
> **(R10).** Thanks for your great suggestions! **We believe there are misunderstandings. We have tested ogbn-arxiv, denoted as Arxiv, in Table 1 in our paper.** We also want to kindly remind the reviewer that we have tested GraphACL on 15 diverse datasets, as listed in Table 4, including various areas, sizes, and homophily ratios.
>
> **(C11). The node clustering results should be included in the main paper**
>
> **(R11).** Thank you! We will move them from the appendix to the main paper.
>
> **We sincerely hope that our responses can address your comments and hope you will consider increasing your score. Thank you so much for your efforts!**
>
> [1] Link Prediction with Non-Contrastive Learning. ICLR 2023
>
> [2] Multi-task Self-supervised Graph Neural Networks Enable Stronger Task Generalization. ICLR 2023

---

> > ### Comment · Reviewer_ycWA · 2023-08-14
> > **Thank you**
> >
> > Thanks for your responses and comprehensive rebuttal.  In particular, I appreciate the experiments and references on low-frequency claims about existing GCL methods, which helps strengthen this work's motivation considerably (I'd make sure you add this in the new version, as it helps this work a lot).
> >
> > R6: Thanks for the clarification.  I missed this experiment and indeed it helps understand that the loss helps but is not required to avoid collapse.  You may want to reference [1] from my response -- this work also introduces some auxiliary augmentations (instead of a uniformity loss term), but for the same effect of helping methods which have such asymmetric predictor structure avoid collapse.  It may help the discussion around why the uniformity loss works in the paper and how it helps the reference GraphACL method achieve better performance.
> >
> > R9: Thanks for adding this.  It strengthens the work and helps support the argument of generality of representations.
> >
> > In light of the above updates, I will raise my score to 6.

---

> > > ### Author Response · Authors · 2023-08-14
> > > **Thanks for your updates and updating the score!**
> > >
> > > Thank you very much for reviewing our paper and reading our rebuttal. We sincerely appreciate your recognition of our clarifications and the increase in your score! In the new version, we will include the experiments and references related to the low-frequency claims about existing GCL methods, as well as cite the work [1] from your response to further illustrate the uniformity loss and its relation to our work.
> > >
> > > Best wishes,
> > >
> > > Authors

---

### Official Review · Reviewer_1K5Q · 2023-07-06

**Soundness:** 3 good
**Presentation:** 3 good
**Contribution:** 3 good
**Rating:** 6
**Confidence:** 4

**Summary:**

This paper propose a simple and effective contrastive learning framework named GraphACL for both homophilic and heterophilic graphs. In particular, GraphACL can capture both one-hop local neighborhood context and two-hop monophily similarties in one single objective. The authors theoretically analyze the learning process of the GraphACL and show that GraphACL can explicitly maximize the mutual information between representations and one-hop neighborhood patterns. The experiments and theoretical analysis show the effectiveness of the proposed approach.

**Strengths:**

-This paper is well written and well structured. The authors first analyze the common limitations of existing GCL frameworks, and then propose some ideas and validate them on many datasets, and finally solve these limitations through the proposed methods and theorems, making this paper well understandable.
-The experiments and theories are sufficient to demonstrate the effectiveness.
-This paper is an early investigation of GCL on heterophilic graphs, which provides a new direction for the subsequent development of GCL.

**Weaknesses:**

-It is common to know about the graph contrastive learning with graph augmentations mentioned in paragraph 3 of the introduction. For the first contrastive scheme mentioned in paragraph 2, the authors should give more details.
-The authors should make a more specific presentation and explanation of the high-frequency and low-frequency information mentioned in the introduction and give a specific instance that high-frequency information is more beneficial for heterophilic graphs.
-Some nodes mentioned in the description of the monophily property of the graphs in lines 190 to 192 can be corresponded to the subgraph d in Figure 3, thus making it less difficult for the reader to read.
-The experimental setup is inconsistent. In the 296 line the authors introduce experiments on five random seeds, while in section 6.2 it is changed to ten runs.

**Questions:**

1.The 184 lines of "node identity and node preference in real-world graphs" is not very understandable. Is this a metaphor of your own?
2.In a general GCL methods (such as GraphCL), the representations are usually mapped into the contrastive space via a projection head before the contrastive loss is calculated. Is this operation similar to the one you mentioned in this paper.
Some typos:
-In Eq.9, $\mathcal{L}_{A C L}(q)$ -> $ \mathcal{L}_{A}(q) $

---

> ### Author Rebuttal · Authors · 2023-08-10
>
> Dear reviewer 1K5Q,  **we appreciate your great summarization and recognition of our contributions and your positive comments on our work: "well written," "early investigation," and "sufficient experiments and theories."** Please find our responses to your comments below:
>
> **(C1). For the first contrastive scheme mentioned in paragraph 2, the authors should give more details.**
>
> **(R1).** Thank you for your valuable suggestion! The first contrastive scheme differs from augmentation-based methods. Instead, it leverages the graph's rich structure to generate contrastive signals [1,2,3,4]. This scheme operates on the homophily assumption, aiming to ensure that connected nodes exhibit similar representations in the latent space. Typically, this scheme utilizes contrastive losses similar to shallow embedding algorithms, as depicted in Table 1 of the related work [4]. We will give and emphasize these details in the revision.
>
> **(C2). The authors should make a more specific presentation and explanation of the high-frequency and low-frequency information mentioned in the introduction and give a specific instance that high-frequency information is more beneficial for heterophilic graphs.**
>
> **(R2).** We greatly appreciate the reviewer's insightful suggestion, which we believe will improve our work. **Please see our detailed responses in Global Response 2**, where we provide a more specific presentation and explanation of the high-frequency and low-frequency.
>
>
> **(C3). Some nodes mentioned  in lines 190 to 192 can be corresponded to the subgraph d in Figure 3.**
>
> **(R3).** Thanks for your suggestion! We will correspond the mentioned nodes to the subgraph d in Figure 3.
>
> **(C4). In the 296 line the authors introduce experiments on five random seeds, while in section 6.2 it is changed to ten runs.**
>
> **(R4).**  **We apologize for your confusion and we believe there are some misunderstandings.** The reason we conducted ten runs is because the standard and public split for heterophilic graphs follows a fixed 10-fold split. On the other hand, for homophilic graphs, the standard and public split is the fixed one-fold split, which is why we just used five random seeds in those experiments. However, to ensure  consistency and robustness, we also ran ten random seeds for the homophilic graphs using our published code. The results  in the table below with ten random seeds showed minimal (no) differences when compared to the five random seeds results. We apologize for your confusion and will more clearly state this in the revision.
>
> | Dataset  |     Cora |     Citeseer  |    Pubmed  | Computer |    Photo|   Arxiv-year |
> | --- | --- | --- | --- | --- | --- | --- |
> | 10 random seeds | 84.20±0.27  | 73.62±0.20 | 82.01±0.13 | 89.83±0.22 | 93.31±0.18 |71.75±0.31 |
> | 5 random seeds | 84.20±0.31  | 73.63±0.22 | 82.02±0.15 | 89.80±0.25 | 93.31±0.19 |71.72±0.26 |
>
>
>
> **(C5). The  "node identity and node preference in real-world graphs" is not very understandable. Is this a metaphor of your own?**
>
> **(R5).** Thanks for your insightful questions! Regarding these concepts, please see our clarifications and responses in **Global Response 1.**
>
>
> **(C6). Is the projection head in GraphCL similar to the one you mentioned in this paper.**
>
> **(R6).** Thanks for your insightful question! The projection head in GraphCL is **inherently different** from our work:
>
> - **Motivation and Method**. The projection head in GraphCL is symmetrical, relies on graph augmentations, and is built based on the idea that augmentations can preserve the semantic nature of samples, i.e., augmented nodes have consistent semantic labels with the same original nodes. However, our predictor is asymmetric and does not rely on augmentations. Instead, GraphACL directly considers predicting the neighborhood distribution in the original graph via an asymmetric predictor. Thus, our motivation and model is inherently different from GraphCL.
>
> - **Theories**. As also noticed by the reviewer, we provide a theoretical analysis to show the connection between our asymmetric predictor and one-hop neighbor context and two-hop monophily, and we prove that learned representations by GraphACL probably enjoy good downstream performance. However, the projection head in GraphCL is unclear theoretically and intuitively in its ability to capture structural information in graphs, especially for heterophilic graphs. This significant difference further sets our work apart from GraphCL.
>
> - **Experiments**. We conducted extensive experiments and analysis on both homophilous and heterophilic graphs. While GraphCL only works on homophilous graphs, for heterophilic graphs, contrastive learning has received little attention to date. Thus, we believe that demonstrating the effectiveness of our GraphACL on heterophilic graphs further distinguishes it from GraphCL
>
> **(C7). Some typos $\mathcal{L}_{ACL} (q) $ in Eq.9.**
>
> **(R7).** Thanks for pointing out the typo! We have corrected it.
>
> **We sincerely hope that our responses can address your comments. Moreover, as noticed by the reviewer, our work presents some interesting findings, a simple yet effective framework, and some theoretical contributions. The reviewer's suggestions can be easily and effectively addressed, and we genuinely hope that the reviewer can consider increasing the score. Please feel free to let us know if there are any remaining comments. Thank you for your efforts!**
>
> [1]  Variational graph auto-encoders. Variational graph auto-encoders. arXiv preprint arXiv:1611.07308, 2016
>
> [2] Contrastive laplacian eigenmaps. NeurIPS 2021
>
> [3] Localized contrastive learning on graphs. arXiv preprint arXiv:2212.04604, 2022
>
> [4] Representation Learning on Graphs: Methods and Applications. arXiv preprint arXiv:1709.05584

---

> > ### Comment · Reviewer_1K5Q · 2023-08-15
> >
> > Thanks for the authors' responses. I am satisfied with the responses that address my concerns. I raise my score to 6.

---

> > > ### Author Response · Authors · 2023-08-15
> > > **Thank you for your comments!**
> > >
> > > Thank you so much for your efforts! We sincerely appreciate the reviewer for checking our responses and the increase in your score!
> > >
> > > Best wishes,
> > >
> > > Authors

---

### Official Review · Reviewer_fMCY · 2023-07-08

**Soundness:** 3 good
**Presentation:** 3 good
**Contribution:** 3 good
**Rating:** 6
**Confidence:** 4

**Summary:**

The paper presents GraphACL, a contrastive learning method for graph representation learning. It aims to address the limitations of current methods that only consider the homophily property of graphs or rely heavily on graph augmentation methods. The authors propose an asymmetric predictor approach where the one-hop local neighborhood context and the two-hop monophily similarity are captured.

**Strengths:**

- The premise of the need for a new contrastive learning method that does not overly rely on graph augmentation methods or the homophily property of graphs is reasonably argued and valid. Evidence from other studies or more data to support this argument could further its strength.

**Weaknesses:**

- The explanation of how GraphACL works and how it captures both the one-hop local neighborhood context and the two-hop monophily similarity might be difficult to comprehend for readers lacking technical background. The use of concrete examples or analogies could strengthen this argument.

- The argument strength could be improved through more detailed explanation of the theoretical analysis of GraphACL, giving more justifications or linking it to established theoretical or empirical studies in the field of contrastive learning.

- The paper's argument that GraphACL is a superior tool when compared to current state-of-the-art graph contrastive learning methods is backed by empirical evidence on both homophilic and heterophilic graphs. However, some rencent proposed method are missing (i.e., [1] and [2])

[1] Spectral Augmentation for Self-Supervised Learning on Graphs

[2] Spectral Feature Augmentation for Graph Contrastive Learning and Beyond

**Questions:**

see weakness

**Limitations:**


Overall, the paper presents an interesting and valid argument. However, to enhance its strength, the author could provide more empirical data, concrete examples, and clearer explanation of the concepts and methodology.

---

> ### Author Rebuttal · Authors · 2023-08-10
>
> Dear reviewer fMCY,  **we appreciate your positive feedback on our paper's soundness, novel insights, and contribution. Please find our detailed responses below:**
>
> **(C1). The explanation of how GraphACL works and how it captures both the one-hop neighborhood context and the two-hop monophily similarity might be difficult to comprehend for readers lacking technical background. The use of concrete examples or analogies could strengthen this argument.**
>
> **(R1).** We thank the reviewer for your good suggestion, which is helpful for the further improvement of our work! We will give a concrete example to help introduce monophily and one-hop local neighborhood context in the introduction part. More specifically, as mentioned by the reviewer ycWA, we will move our Figure 3 to the beginning of the introduction part, which can serve as a good concrete toy example of various design motivations (homophily, one-hop neighborhood context, and two-hop monophily) of GraphACL.
>
> **(C2). The argument strength could be improved through explanation of the theoretical analysis of GraphACL.**
>
> **(R2).** Thank you for your suggestion. We would like to give the following explanation of our theoretical analysis.  We will highlight the following discussions in the revision.
>
> - (1) Although there are many theoretical works [3,4,5,6,7] trying to theoretically understand how contrastive learning works, these works mainly focus on contrastive learning in the image IID setting, the theoretical analysis on the non-IID node representation learning (each instance is a node in one large graph and instances are inter-connected resulting in a non-IID nature) is still quite limited. Our theoretical analysis shows that our simple GraphACL can simultaneously capture one-hop neighborhood context and two-hop monophily. We also theoretically and empirically show that the learned representations by GraphACL can achieve better downstream performance and the connection between GraphACL and graph eigenvalues. **Thus, our theoretical analysis is significantly different from those works and is complementary to established theoretical studies in contrastive learning.**
>
> - (2) In addition, many established theoretical or empirical studies on image contrastive learning [3,4,5,6,7], and graph contrastive learning [1,2,8,9,10] are based on augmentations and hope that augmentations can preserve the invariant semantic nature of samples, i.e., the augmented samples have invariant semantic labels with the original ones. **However, our theoretical analysis is not based on augmentations but directly considers analyzing neighborhood distribution. Thus, GraphACL provides new perspectives on graph contrastive learning that are quite distinctive from current theoretical and empirical works.**
>
> **(C3). The paper's argument that GraphACL  is a superior tool when compared to current state-of-the-art graph contrastive learning methods is backed by empirical evidence on both homophilic and heterophilic graphs. However, some recent proposed method are missing (i.e., [1] and [2]).**
>
> **(R3).** We thank the reviewer for sharing these two works [1,2]! **We kindly want to remind the reviewer that we have conducted an extensive evaluation on  10 methods on 15 datasets , including many state-of-the-art contrastive learning methods. Thus, we believe our experiments can strongly corroborate the effectiveness and scalability of GraphACL. Nevertheless, we completely agree with the reviewer that including an empirical comparison with SPAN [1] and SFA [2] would be beneficial.** Since the code of SFA [2] has not been publicly available, we decided to compare our GraphACL with COSTA [11], which also employs feature augmentation like SPAN [2] and provides publicly available code. We ran additional experiments on SPAN [1] and COSTA [11]. For all methods and datasets, we used the same public and standard splits as in our paper, and we will include the following results in the revision:
>
> | Dataset  |     Citeseer |  (Ogbn)-Arxiv  |   Squirrel  | Chameleon |     Crocodile|    Arxiv-year |
> | --- | --- | --- | --- | --- | --- | --- |
> | COSTA  | 72.31±0.27 |  71.00±0.40 | 48.36±0.25 | 61.82±0.24 | 59.70±0.33 | 42.21±0.28 |
> | SPAN  | 72.01±0.58  | 70.94±0.24 | 49.47±0.39 | 62.55±0.36 | 61.53±0.52 | 43.95±0.41 |
> | GraphACL | **73.63±0.22**  | **71.72±0.26** | **54.05±0.13** | **69.12±0.24** | **66.17±0.24** |**47.21±0.39** |
>
> The results are encouraging and further strengthen our contribution! The results show that our GraphACL outperforms these two competitive baselines [1,11] based on augmentations, especially on heterophilic graphs.
>
> **In light of these responses, we sincerely hope our rebuttal has addressed your comments, and believe that your comments do not affect our key contributions and can be easily addressed in the revision. We also genuinely hope you will reconsider increasing your score. If you have any other comments, please do share them with us, and we will address them further. Thank you for your efforts!**
>
> [1] Spectral Augmentation for Self-Supervised Learning on Graphs. ICLR 2023
>
> [2] Spectral Feature Augmentation for Graph Contrastive Learning and Beyond. AAAI 2023
>
> [3] A Theoretical Analysis of Contrastive Unsupervised Representation Learning. ICML 2019
>
> [4] Understanding Contrastive Representation Learning through Alignment and Uniformity on the Hypersphere. ICML 2020
>
> [5] Understanding Self-Supervised Learning Dynamics without Contrastive Pairs. ICML 2021
>
> [6] Provable Guarantees for Self-supervised Deep Learning with Spectral Contrastive Loss. NeurIPS 2021
>
> [7] Generalization Analysis for Contrastive Representation Learning. ICML 2023
>
> [8] Graph Contrastive Learning with Augmentations. NeurIPS 2020
>
> [9] Graph Contrastive Learning with Adaptive Augmentation. WWW 2021
>
> [10] Large-Scale Representation Learning on Graphs via Bootstrapping. ICLR 2021
>
> [11] COSTA: Covariance-Preserving Feature Augmentation for Graph Contrastive Learning. KDD 2022

---

> > ### Comment · Reviewer_fMCY · 2023-08-13
> >
> > I am satisfied with author's rebuttal. It would be good to see the new result are included in the revision. I rasie my score from 5 to 6

---

> > > ### Author Response · Authors · 2023-08-13
> > > **Thank you for your comments and updating the score!**
> > >
> > > Thank you very much for carefully reading our response and increasing your score! We are glad our response has addressed your comments. We genuinely appreciate your support and will include the new results in the main paper.
> > >
> > > Best wishes,
> > >
> > > Authors

---

### Author Rebuttal · Authors · 2023-08-10

**We sincerely thank all the reviewers for their insightful comments and helpful suggestions. Overall, the reviewers praised our work's originality, soundness, and clarity. We deeply appreciate the numerous positive comments on our work, such as describing it as "simple, effective, and intuitive," "well-written," and the "solid theoretical and empirical analysis".**

We provide this **Global Response** to address similar comments or misunderstandings from reviewers. Additionally, we have **attached a one-page PDF** containing additional experimental results suggested by some reviewers.

**(Global C1) The concept of  "node identity" and "node preference" (context) in real-world graphs**

**(Global R1)** Thanks for your comments! We would like to clarify this and will add the following responses in the revision.**

The "node true identity" and "node preference" in real-world graphs have been mainly introduced as the social concepts by [1,2]. Specifically, monophily in graphs implies the existence of nodes with strong preferences for specific nodes, which differ from their own identity. For example, users of a certain gender may not always prefer neighbors of the same gender, leading to a preference that deviates from their identity.

In our work, **we incorporate these social concepts into a practical contrastive learning framework.** We achieve this by converting the central node's identity representation to its preference (context) representation using a simple asymmetric predictor. Subsequently, the preference (context) representation of the central node is used to predict identity representations of its neighbors. **Our theoretical and empirical analysis demonstrate GraphACL effectively captures these social concepts, leading to a good performance on both homophilic and heterophilic graphs.**

**(Global C2) More explanation of the high-frequency and low-frequency information for augmentation-based GCL approaches on heterophilic graphs**

**(Global R2)**  Thanks for your great comments! We want to clarify the following points and will add the discussion below in the final version.

- **(1)** From a spectral perspective of graphs, capturing low-frequency signals involves smoothing features across the graph, ensuring similarity among node representations. This is crucial for homophilic graphs. For instance, studies [3,4,5] have shown that using different signals is an effective approach for dealing with diverse graphs: low-frequency signals for homophilic graphs and high-frequency signals for heterophilic graphs
- **(2)** GCL with augmentation operates on the principle that augmentation should retain crucial information, encouraging the model to learn invariant representations by disregarding perturbations in unimportant information. Thus, preserving important frequency signals in graph augmentation becomes essential for GCL. The previous work [6] (i..e, [22] in our paper) has theoretically shown learned representations by GCL with augmentations essentially capturing the invariant low-frequency information. **Consequently, current GCL with augmentation implicitly relies on the homophily assumption, and can not perform well on heterophilic graphs with diverse neighbors [4]. This has been also supported empirically in previous research [4], as well as in our own experimental results on both real-world  and synthetic datasets.**
- **(3) To further support the above motivation and make our work more self-contained, we evaluate the eigenvalue (spectrum) change of the augmented graph views under different methods: GCA [9], BGRL [10], and AD-GCL [11].** The results is shown in Figure 1 in our attached one-page PDF.  The results show that the variation in the low-frequency components are smaller than those in the high-frequency components, for both homophilic and heterophilic graphs. This observation confirms that mainstream GCL methods with augmentations often maintain the invariance of low-frequency signals while perturbing high-frequency signals. Moreover, these methods exhibit lower effectiveness on heterophilic graphs compared to GraphACL, particularly when the perturbation in high-frequency signals is more significant. This highlights the importance of preserving high-frequency signals in augmentation-based GCL methods for heterophilic graphs.

**(Global C3) There are a few approaches missing from baselines, e.g. [12] or [13]**

**(Global R3)** Thanks for raising great works T-BGRL  [12] and PARETOGNN [13]! Following your suggestions, we compare GraphACL with [12,13]. We follow the same data splits as our paper and present the results in Table 1 in our attached one-page PDF. We will include this comparison in our final revision. From the results, we can observe that GraphACL performs better than [12, 13], especially on heterophilic graphs. **These additional results, combined with the comparison against 10 baselines on 15 datasets using our source code in our submission, can strongly validate the effectiveness of GraphACL.**

[1] Monophily in Social Networks Introduces Similarity among Friends-of-friends. Nature Human Behaviour. 2018.
[2] Decoupled Smoothing on Graphs. WWW 2019. [3] Beyond Low-frequency Information in Graph Convolutional Networks. AAAI 2021.
[4] Adaptive Universal Generalized PageRank Graph Neural Network. ICLR 2021.
[5] Revisiting Heterophily For Graph Neural Networks. NeurIPS 2022.
[6] Revisiting Graph Contrastive Learning from the Perspective of Graph Spectrum. NeurIPS 2022.
[7] Decoupled Self-supervised Learning for Graphs. NeurIPS 2022.
[8] Deep Graph Infomax. ICLR 2019.
[9] Graph Contrastive Learning with Adaptive Augmentation. WWW 2021.
[10] Large-Scale Representation Learning on Graphs via Bootstrapping. ICLR 2022.
[11] Adversarial Graph Augmentation to Improve Graph Contrastive Learning. NeurIPS 2021.
[12] Link Prediction with Non-Contrastive Learning. ICLR 2023.
[13] Multi-task Self-supervised Graph Neural Networks Enable Stronger Task Generalization. ICLR 2023

---

### Decision · Program_Chairs · 2023-09-21

**Decision:**

Accept (poster)

**Comment:**

The reviewers' consensus is that the paper is well-written, and the claims are supported both by experimental evidence and theoretical analysis.